# Multivalent interactions with CCR4–NOT and PABPC1 determine mRNA repression efficiency by tristetraprolin

Filip Pekovic [1], Wi S. Lai [2], Joshua Corbo [1], Stephanie N. Hicks [2], Keiko Luke[1], Perry J. Blackshear [2,3] ✉ & Eugene Valkov [1] ✉

Tristetraprolin family of proteins regulate mRNA stability by binding to specific AU-rich elements in transcripts. This binding promotes the shortening of the mRNA poly(A) tail, or deadenylation, initiating mRNA degradation. The CCR4–NOT complex plays a central role in deadenylation, while the cytoplasmic poly(A)-binding protein PABPC1 typically protects mRNAs from decay. Here, we investigate how tristetraprolin interacts with CCR4–NOT and PABPC1 to control mRNA stability. Using purified proteins and in vitro assays, we find that tristetraprolin engages CCR4–NOT through multiple interaction sites and promotes its activity, emphasizing the importance of multivalent binding for efficient deadenylation. Phosphorylation of tristetraprolin does not affect its interaction with CCR4–NOT or its deadenylation activity, but is essential for tristetraprolin's binding to PABPC1. We propose that tristetraprolin promotes the processive deadenylation activity of CCR4–NOT on mRNAs containing AU-rich elements, with phosphorylation-dependent interactions with PABPC1 potentially enhancing deadenylation and promoting regulated mRNA decay.

Tristetraprolin (TTP), also known as zinc finger protein 36 (ZFP36), is a prototypical member of a family of RNA-binding proteins that regulate mRNA stability in eukaryotic cells[1–4]. TTP binds to AU-rich elements (AREs), specifically motifs of the UUAUUUAUU type, within the 3′ untranslated regions (UTRs) of target mRNAs through its tandem zinc finger (TZF) domain. These AREs are prevalent in mRNAs encoding cytokines, growth factors, and proto-oncogenes and are associated with rapid mRNA turnover[5,6]. Upon binding to AREs, TTP promotes the shortening of the poly(A) tail[7,8] — a process known as deadenylation — which is the initial and rate-limiting step in mRNA degradation in eukaryotes[9] and is essential for controlling gene expression levels[10,11].

The critical role of TTP in regulating inflammatory responses is underscored by observations that loss of its expression or disruption of its RNA-binding ability leads to the accumulation of pro-inflammatory cytokines, such as tumor necrosis factor α (TNFα), leading to pathological conditions including systemic inflammation[12,13] and cancer[14,15].

Notably, elevated levels of human antigen R (HuR), another ARE-binding protein that stabilizes TTP's target mRNAs by competing for the same AREs[16], also contribute to inflammation[17]. These observations highlight the essential function of TTP in controlling the expression of inflammatory cytokines through mRNA destabilization.

Multiple lines of evidence suggest that TTP exerts its mRNA-destabilizing function by interacting with CCR4-NOT, a multi-protein complex that is the principal deadenylase in eukaryotic cells[18–20]. A conserved C-terminal peptide in TTP, known as the C-terminal NOT1-binding domain (CNBD), has been shown to form an amphipathic α-helix that directly interacts with a HEAT repeat-containing domain of the NOT1 subunit of CCR4–NOT[2,21]. However, deletion of the CNBD results in only a modest reduction in TTP's ability to promote mRNA decay in cell-free assays, HEK-293 cells and in knock-in mouse models[21,22], suggesting that additional regions within TTP contribute to CCR4–NOT recruitment and function.

[1]National Cancer Institute, National Institutes of Health, Frederick, MD 21702, USA. [2]National Institute of Environmental Health Sciences, National Institutes of Health, Research Triangle Park, Durham, NC 27709, USA. [3]Departments of Medicine and Biochemistry, Duke University Medical Center, Durham, NC 27710, USA. ✉e-mail: black009@niehs.nih.gov; eugene.valkov@nih.gov

Further supporting this notion, studies in *Schizosaccharomyces pombe* have shown that Zfs1p, the single TTP family member in this organism, can interact with CCR4−NOT despite lacking a CNBD analogous to that of human TTP[23]. Similar observations have been made in other non-mammalian organisms, where deletion of the C-terminal domain in TTP orthologs had varying effects on mRNA decay activity[24,25]. These findings support the idea that alternative or additional domains within TTP may be involved in CCR4−NOT interactions and mRNA regulation.

Phosphorylation is another factor that may influence TTP function. TTP and its related proteins are extensively phosphorylated[26,27], which may modulate their activity, stability, and interactions[28–31]. Specifically, phosphorylation at serine residues 60, 186, and 323 in human TTP (corresponding to serines 52, 178, and 316 in mouse TTP) has been reported to reduce TTP-dependent mRNA decay[32–34]. Furthermore, it was reported that phosphorylation of TTP decreases its interaction with CCR4−NOT[33–35]. However, the role of phosphorylation in regulating TTP's interactions with CCR4−NOT and its overall activity remains controversial.

In addition to CCR4−NOT, TTP has been reported to interact with the cytoplasmic poly(A)-binding protein PABPC1[36,37]. PABPC1 typically binds to the poly(A) tail of mRNAs, protecting them from deadenylation and enhancing their stability[38]. The interaction between TTP and PABPC1 is thought to influence mRNA decay and translation, although the functional consequences are not fully understood. Some studies suggest that the TTP:PABPC1 interaction interferes with TTP-stimulated deadenylation[39], while others propose that it plays a role in TTP-mediated translational repression[36].

In this study, we identify multiple regions within human TTP that are critical for its interaction with the human CCR4−NOT, demonstrating that multivalent interactions contribute to efficient, ARE-specific deadenylation. In cells, this activity leads to repression, which we define as CCR4−NOT-mediated deadenylation-dependent decay of target transcripts leading to silencing of translational output.

We show that the phosphorylation status of TTP does not impair its association with CCR4−NOT or its ability to promote mRNA deadenylation or decay, but is essential for its interaction with PABPC1. Furthermore, we reveal that TTP's family members, ZFP36L1 and ZFP36L2, exhibit similar interaction patterns with CCR4−NOT and PABPC1, and that phosphorylation influences these interactions in a conserved manner. Our findings provide comprehensive insights into the molecular mechanisms by which TTP and its family members regulate mRNA stability, advancing our understanding of post-transcriptional gene regulation and highlighting potential avenues for therapeutic intervention in diseases associated with dysregulation of mRNA stability.

## Results

### Multiple TTP interfaces compensate for loss of canonical CCR4−NOT binding sites

TTP and its related proteins bind AU-rich elements (AREs) in mRNA through their tandem zinc finger (TZF) domain[4,40], comprising two CCCH zinc fingers with defined spacing, flanked by two low-complexity regions (Fig. 1a). The C-terminal 13 residues (CNBD) of human TTP were previously shown to interact with a central domain of the NOT1 subunit of CCR4−NOT[21]. The CNBD is conserved in many eukaryotes, including primitive plants. However, many eukaryotes, such as most fungi and many higher plants, lack a characteristic CNBD[2] (Supplementary Fig. 1). Notably, despite lacking a defined CNBD, the *Schizosaccharomyces pombe* TTP family member Zfs1p can efficiently stimulate rapid targeted deadenylation by *S. pombe* CCR4−NOT in vitro[23]. Deleting the CNBD in human TTP only modestly reduced mRNA decay and deadenylation in HEK-293 cell transfections and complementary in vitro assays using recombinant *S. pombe* CCR4−NOT[22,41]. In mice, deletion of the corresponding TTP sequence

produced a milder inflammatory phenotype than a complete TTP knockout[22].

We confirmed these results in vitro using a fully reconstituted recombinant human CCR4−NOT complex[42], consisting of the seven subunits NOT1, NOT2, NOT3, NOT6, NOT7, NOT9, NOT10, and NOT11, and purified TTP proteins. Pull-down assays showed that the CNBD deletion did not substantially reduce TTP binding to CCR4−NOT (Fig. 1b). In vitro, full-length TTP (FL) robustly drove ARE-targeted deadenylation of a synthetic fluorescently-labeled RNA containing a single UUAUUUAUU motif and a 20-nucleotide poly(A) tail (Fig. 1c). Surprisingly, TTP lacking the CNBD (ΔCNBD) stimulated deadenylation almost as effectively as the full-length protein. A point mutation in TTP's RNA-recognition site (C124R) that completely prevents RNA binding[7] confirmed that TTP tethers CCR4−NOT to RNA rather than enhancing CCR4−NOT's intrinsic activity. These results suggested that TTP might have multiple CCR4-NOT interaction sites beyond the CNBD, supporting previous findings[18,19,41].

Four tryptophan residues in human TTP were previously demonstrated to be involved in TTP's interaction with CCR4−NOT's NOT9 subunit (Fig. 1a, d)[18]. However, because these residues are poorly conserved across species and are entirely absent in the human TTP family members ZFP36L1 and ZFP36L2 (Supplementary Fig. 2), their universal role in CCR4−NOT interactions remains uncertain. Our direct interaction studies showed that mutating all four tryptophans to alanine in full-length human TTP eliminated its interaction with NOT9 (Supplementary Fig. 3), while any single tryptophan mutation maintained it. However, the TTP tryptophan mutant with all four tryptophans mutated to alanines (4xW/A) did not show reduced targeted deadenylation stimulation (Fig. 1e), suggesting that other regions within TTP may compensate for the lost putative NOT9 interaction.

To assess the cellular impact of the CNBD deletion and tryptophan mutations, we measured interleukin-3 (IL-3) production, a well-established TTP target transcript[43–45], through cell-based co-transfection assays (Fig. 1f, g). The CNBD deletion and tryptophan mutations resulted in only modest relief of IL-3 repression. These findings align with our in vitro deadenylation and interaction data, supporting the idea that multiple TTP regions are involved in CCR4−NOT recruitment and ARE-specific deadenylation.

### TTP exploits its intrinsically disordered regions to recruit CCR4−NOT

We comprehensively characterized TTP:CCR4−NOT interactions by using fully reconstituted human CCR4-NOT and its subcomplexes to assess all modules, rather than focusing on individual components as in previous studies (Fig. 2a)[42]. Direct interaction assays demonstrated that full-length human TTP binds stoichiometrically to the central HEAT domain of NOT1 and sub-stoichiometrically to the NOT9 module (Fig. 2b). We also discovered a stoichiometric interaction between TTP and the NOT module, which comprises the C-terminal Spa2 Homology Domain (SHD) together with NOT2 and NOT3, thereby defining three distinct CCR4−NOT binding interfaces.

We further validated the interaction with the NOT module in HEK-293T cells via a colocalization assay[46] with fluorescent tags (Supplementary Fig. 4a, b). Using EGFP-NOT1 SHD alongside PH-mCherry-TTP (which carries a pleckstrin homology domain to anchor TTP at the plasma membrane), we first observed tight colocalization of NOT1 SHD with TTP at the cell periphery. To test competition with established NOT module interactors, we next co-expressed OST4-mCherry-NANOS3 (anchoring the NANOS3 fragment to the endoplasmic reticulum) with EGFP-NOT1 SHD, which produced a characteristic perinuclear ring of overlap at the endoplasmic reticulum. Strikingly, when PH-mCherry-TTP was also present, EGFP-NOT1 SHD relocated exclusively to the plasma membrane with TTP, while the OST4-mCherry-NANOS3 signal became diffuse around the nucleus, mirroring negative

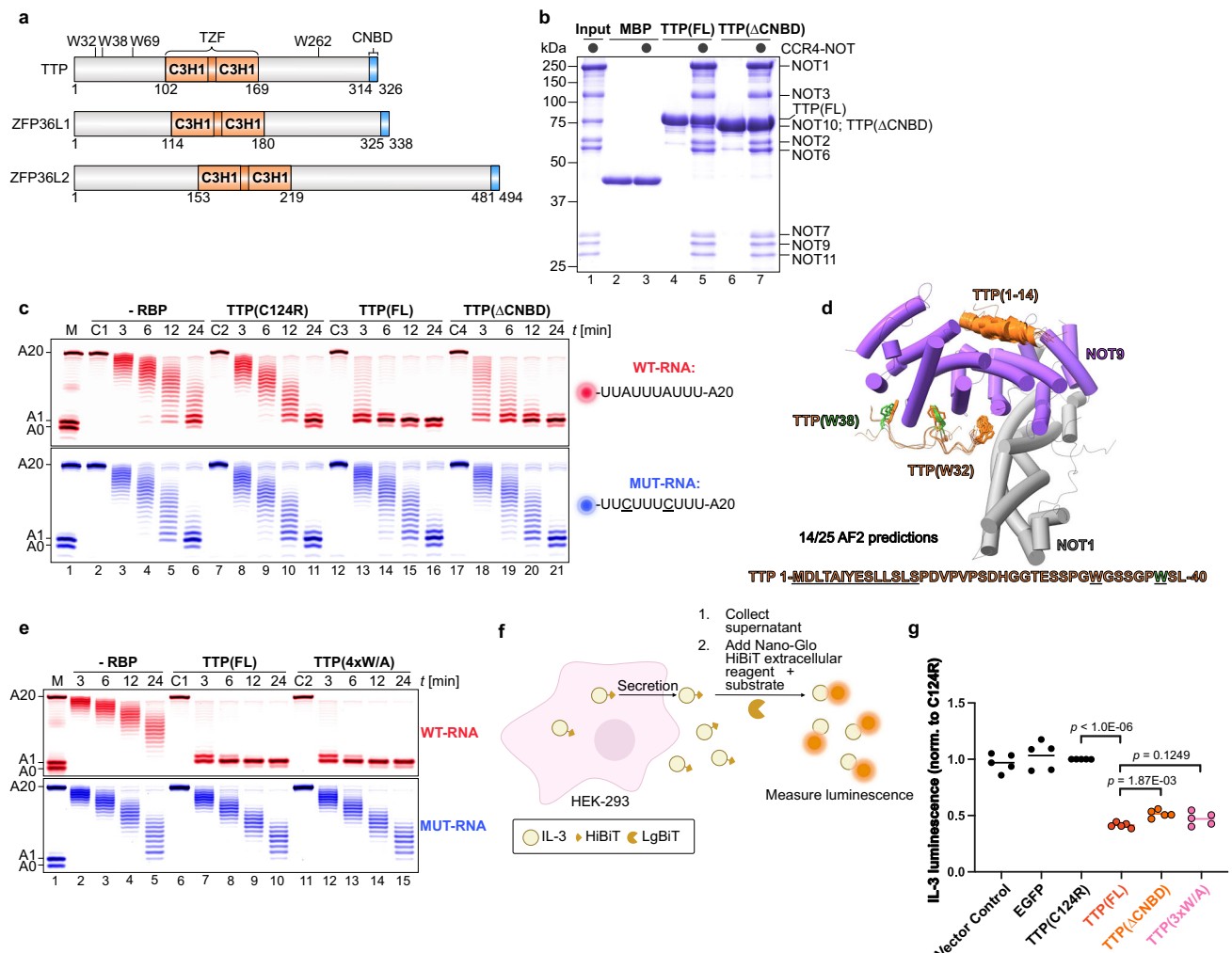

**Fig. 1 | Assessing the role of the C-terminal motif and tryptophan residues in TTP for CCR4–NOT recruitment and repression. a** Schematic representation of human tristetraprolin (TTP) and its human family members ZFP36L1 and ZFP36L2, highlighting the tandem zinc finger (TZF) domain necessary for RNA binding, containing two C3H1 zinc fingers, and the C-terminal NOT1-binding domain (CNBD). Also noted are the positions of the key tryptophan residues in TTP. **b** Pull-down assays comparing the interaction between CCR4–NOT (black circles) and either full-length TTP (FL) or TTP lacking the CNBD (ΔCNBD). Maltose-binding protein (MBP) was included as a negative control. Here and in all other figures, proteins were resolved by SDS-PAGE and visualized by Coomassie staining. **c** In vitro deadenylation assays demonstrating ARE-specific (WT-RNA, red, containing one TTP-binding motif; mutant RNA (MUT-RNA), blue) and TTP-dependent stimulation of deadenylation, compared with ΔCNBD and a non-RNA-binding mutant of TTP (C124R). Substrate RNAs (50 nM each) were incubated with 50 nM CCR4–NOT in the presence or absence (-RBP) of 100 nM TTP protein. Control reactions included RNA alone (C1) and RNA with the indicated TTP variants (C2–C4). Here and in all other figures, RNAs were resolved by urea-PAGE and

visualized by fluorescence detection. The first lane here and in all following dead-enylation assay gels indicates the size marker corresponding to the number of adenosines present. **d** AlphaFold2 predictions depicting fourteen out of twenty-five models of TTP interacting with the NOT9 module of CCR4–NOT. The first fourteen residues of TTP (orange) are predicted to bind the concave surface of NOT9 (purple), with tryptophan residues W32 (orange) and W38 (green) positioned near the tryptophan-binding pockets of NOT9. (**e**) Deadenylation assays as in (**c**), comparing full-length TTP (FL) to the TTP variant in which all tryptophan residues were mutated to alanine (4 × W/A). **f** Schematic illustration of the Nano-Glo HiBiT extracellular assay used to measure IL-3 protein production. Prepared in Affinity Designer 2. **g** Repression of IL-3 protein production by TTP variants. IL-3 lumines-cence was normalized to levels observed in cells expressing the non-RNA-binding mutant TTP(C124R). TTP(3×W/A) had tryptophan to alanine mutations at positions 32, 38, and 69. Statistical significance was determined using a two-tailed Student's *t*-test followed by Holm-Šidák correction for multiple comparison and shown as exact *p* values. Each data point represents one biological replicate (*n* = 5).

controls (NANOS3 with EGFP alone or OST4-mCherry with NOT1 SHD). These results indicate that TTP efficiently outcompetes the NANOS3 fragment, which interacts with NOT1 via a single α-helix[47], for binding to NOT1 SHD.

To delineate the minimal CCR4–NOT architecture required for TTP-driven deadenylation, we reconstituted four human sub-complexes: FULL (all three TTP-interaction regions), MINI (two regions), CORE (one region), and NOT6:7 (no regions), and assayed their activity (Fig. 2c). Only subcomplexes containing the NOT module – the presently identified TTP binding interface – supported complete, TTP-stimulated deadenylation, whereas the CORE and NOT6:7

assemblies failed to do so. Remarkably, the MINI complex, despite lacking the canonical NOT1 HEAT-domain TTP contact[21], recapitulated the full deadenylation activity of the intact complex. These findings demonstrate that engagement of the NOT module is necessary for efficient TTP-mediated deadenylation and establish this interface as a central mechanistic determinant.

Further analyses with fragments of TTP revealed that TTP's intrinsically disordered regions (IDRs), located both N- and C-terminal to the RNA-binding TZF domain, support interactions with CCR4–NOT (Fig. 2d). The C-terminal IDR exhibited strong interac-tions with the extended NOT1:10:11 and NOT modules and weak

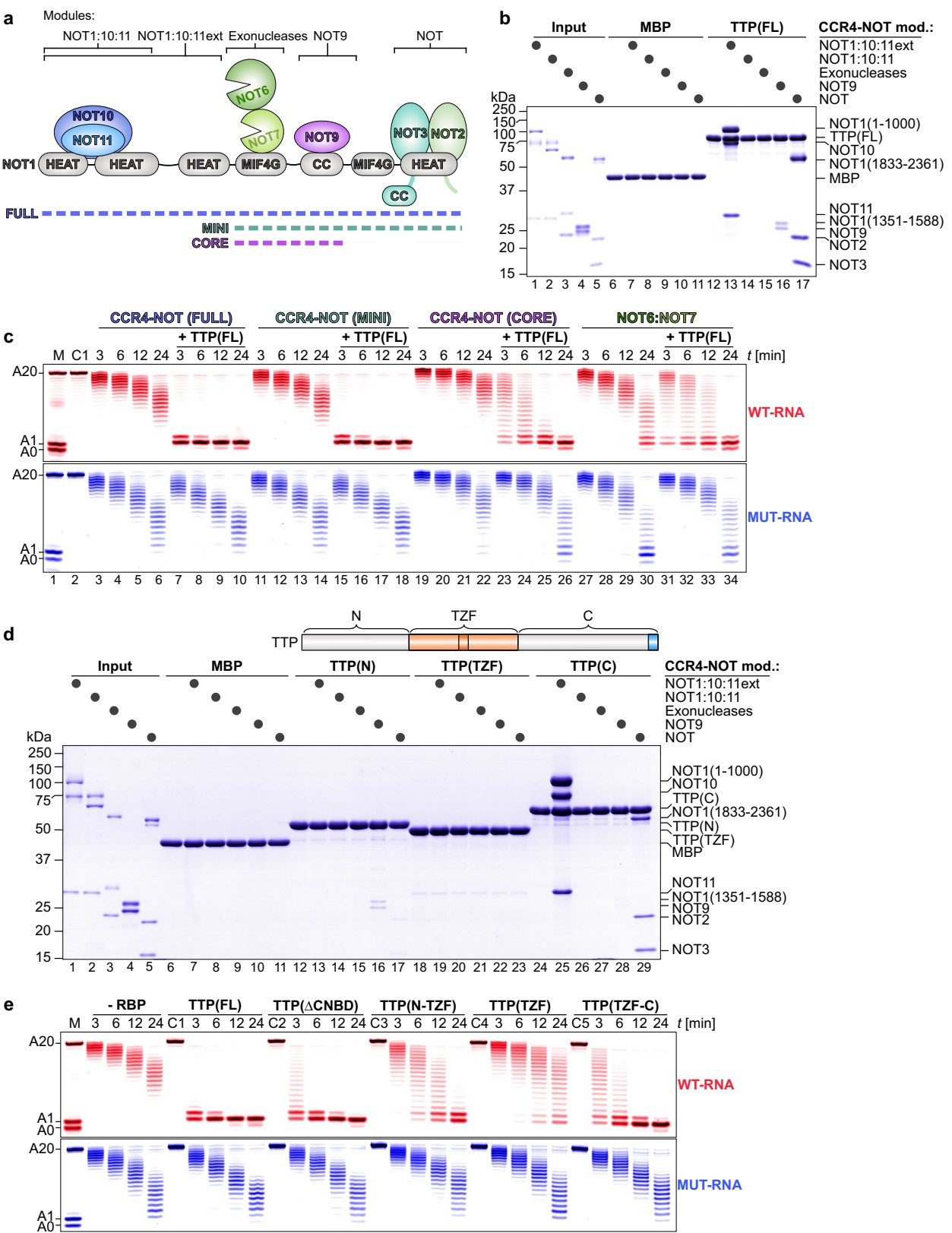

interactions with the NOT9 module. In contrast, the N-terminal IDR showed only weak, sub-stoichiometric interactions with the NOT9 and NOT modules.

To define which TTP regions are required for CCR4−NOT-mediated deadenylation, we tested the full complex with either full-length TTP, the TZF domain alone, or TZF fused to the N- or C-terminal IDRs

(Fig. 2e). The isolated TZF domain was inactive, whereas both IDR-TZF fusions restored targeted deadenylation, with the C-terminal IDR fusion driving the most rapid poly(A) shortening. This stimulatory hierarchy parallels the binding pattern observed in pull-down assays (Fig. 2d) and is consistent with studies in *S. pombe*, where low-complexity regions similarly enhance CCR4−NOT function[23].

**Fig. 2 | TTP requires multiple CCR4–NOT interactions for efficient targeted deadenylation. a** Schematic representation of the human CCR4–NOT complex architecture, indicating the subcomplexes used: MINI (lacking the N-terminal portion of NOT1 and the NOT10/11 proteins) and CORE (lacking both the N- and C-terminal portions of NOT1 and the NOT2/3/10/11 proteins). Domains of NOT1 are denoted as follows: HEAT domain (α-helical HEAT-like repeats), MIF4G domain ('middle of 4G'), and coiled-coil (CC) domain. Prepared in Affinity Designer 2. **b** Pull-down assays showing the interaction of full-length TTP (FL) with the indicated CCR4–NOT modules (black circles). MBP was included as a negative control. **c** Deadenylation assays comparing targeted deadenylation by full-length TTP (TTP-FL, 100 nM) in the presence of different CCR4–NOT subcomplexes. All CCR4–NOT subcomplexes were added at equimolar concentrations (50 nM) relative to

substrate RNAs (50 nM), except for the exonuclease heterodimer, which was used at 250 nM. A control reaction containing only RNA and TTP (C1) was included. **d** Pull-down assays demonstrate that TTP's N-terminal and C-terminal intrinsically disordered regions (IDRs) interact with multiple CCR4–NOT modules, as indicated (black circles). MBP was included as a negative control. The domain architecture of full-length TTP highlighting the N- and C-terminal IDRs surrounding the TZF domain is given for reference. The CNBD is included in the C-terminal IDR fragment. **e** Deadenylation assays comparing the stimulation of deadenylation by TTP (FL) with ΔCNDB and TTP variants in which either IDR was fused to the TZF RNA-binding domain. All TTP variants were added at 100 nM, in twofold excess over CCR4–NOT (50 nM) and substrate RNAs (50 nM). Controls included reactions containing only RNA and the indicated TTP variant (C1-C5).

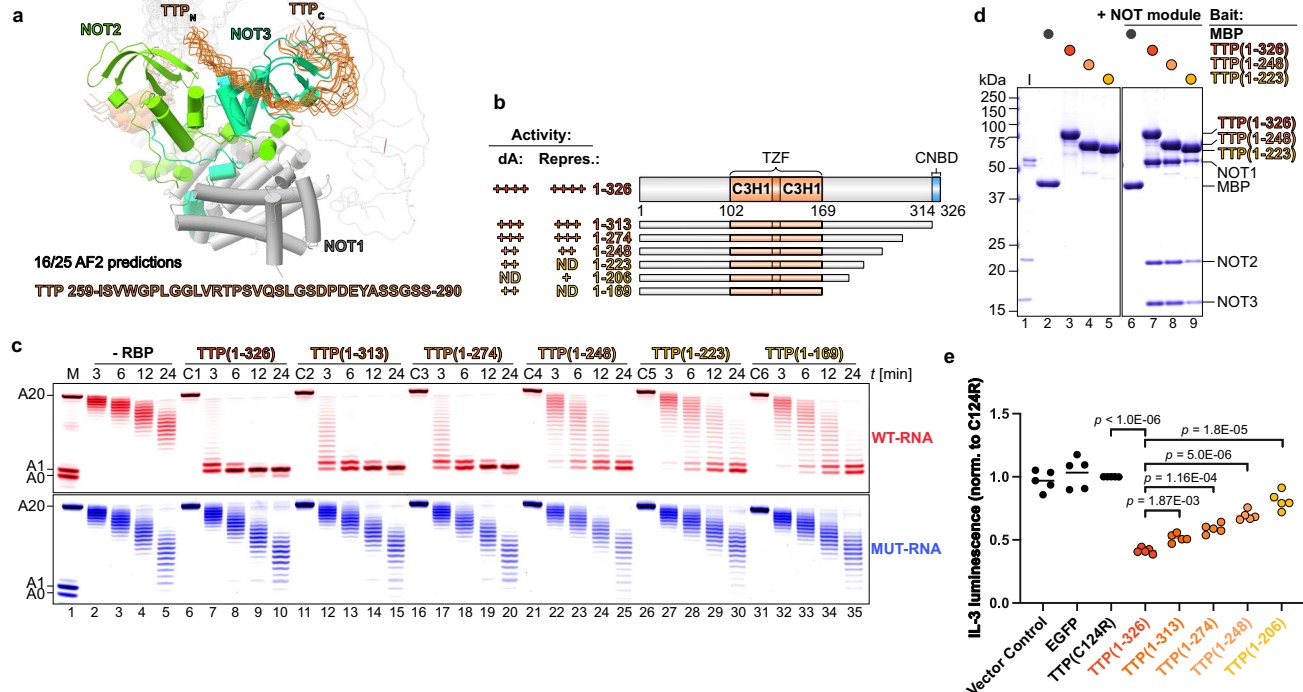

**Fig. 3 | The C-terminal unstructured region of TTP is essential for interaction with the NOT module and efficient deadenylation. a** Alignment of sixteen out of twenty-five AlphaFold2 predictions reveals a potential interaction surface between TTP (orange) and the NOT module (NOT1 in grey, NOT2 in light green, NOT3 in dark green). A short region within the C-terminal IDR of TTP converges around NOT3. **b** Domain organization and summary of activity for the C-terminal truncations of TTP used in deadenylation assays (labeled 'dA'; panel c) and protein production assays (labeled 'Repres.'; panel e). **c** Deadenylation assays assessing the effect of gradual C-terminal truncations of TTP on targeted deadenylation. All TTP variants

were added at 100 nM, in twofold excess over CCR4–NOT (50 nM) and substrate RNAs (50 nM). Control reactions included reactions with RNA and the indicated TTP variant (C1-C6) alone. **d** Pull-down assays with the NOT module and the indicated TTP fragments (colored circles). MBP (black circle) was included as a negative control. **e** Effect of C-terminal IDR truncations of TTP on the repression of IL-3 protein production in co-transfection assays, measured by relative luminescence. Statistical significance was determined using a two-tailed Student's *t*-test followed by Holm-Šidák correction for multiple comparison and shown as exact *p* values. Each data point represents one biological replicate (*n* = 5).

## NOT module interaction is required but insufficient for full TTP activity

Two lines of evidence pointed to the NOT module as the key driver of rapid TTP-stimulated deadenylation: first, the NOT module binds strongly to TTP's C-terminal IDR, and second, CCR4–NOT subcomplexes containing the NOT module exhibit the greatest stimulation by TTP. Because deletion of the N-terminal IDR or disruption of either the NOT9 or NOT1 contacts had little effect on deadenylation rates, we therefore turned our attention to the C-terminal IDR. We sought to pinpoint the minimal segment within this region responsible for engaging the NOT module, reasoning that its interaction might underlie TTP's ability to markedly accelerate deadenylation.

To identify the specific region of TTP's C-terminal IDR interacting with the NOT module, we used AlphaFold2 to predict the complex between the NOT module and TTP's C-terminal IDR (Fig. 3a). This

revealed a 32-residue region that wraps around the NOT3 subunit. Based on this prediction and the sequence conservation within the C-terminal IDR (Supplementary Fig. 1), we designed a series of TTP C-terminal truncations (Fig. 3b). We then tested these C-terminal truncations in deadenylation assays, measured their effect on IL-3 protein production in cells, and assessed their ability to bind directly to the NOT and NOT9 modules (Fig. 3b–e).

Truncation of the C-terminal 78 residues (TTP 1–248) markedly impaired targeted deadenylation compared to either full-length TTP 1–326 or the CNBD-deleted variant TTP 1–313 (Fig. 3b, c). Despite retaining near-wild-type binding to the NOT module (Fig. 3d), TTP 1–248 lost strong association with NOT9 (Supplementary Fig. 5) and, owing to its CNBD removal, lacked NOT1 engagement. Shorter C-terminal deletions did not further reduce deadenylation yet progressively weakened NOT module binding (Fig. 3d). Moreover,

successive elimination of individual CCR4−NOT interaction surfaces within the C-terminal IDR produced a stepwise relief of IL-3 repression in cells (Fig. 3e). Together, these data indicate that although binding of the NOT module is necessary, it alone is insufficient for rapid deadenylation while remaining important for efficient post-transcriptional regulation in cells.

## Multivalent TTP C-terminal IDR binding to CCR4−NOT drives rapid deadenylation

Next, we sought to identify the specific regions within TTP's C-terminal IDR essential for rapidly accelerating deadenylation. Initial observations showed that the NOT module strongly interacts with the C-terminal IDR and is part of CCR4−NOT subcomplexes most stimulated by TTP. However, this interaction alone proved insufficient for rapid deadenylation, suggesting the involvement of at least one additional CCR4−NOT module.

We identified a 50-residue region of TTP between residues 224 and 274 that was critical for deadenylation stimulation and interactions with the NOT and NOT9 modules (Fig. 4a−c, Supplementary Fig. 6a−d). However, this region alone as the sole CCR4−NOT interaction surface was insufficient for rapid deadenylation acceleration. Comparing TTP fragments 102−326 and 102−313 (which lacks the CNBD), we observed similar binding levels to the NOT and NOT9 modules (Supplementary Fig. 6a, b). Despite this, both fragments show strongly decreased NOT9 module binding compared to the full-length protein.

A TTP fragment lacking residues 224−274 (Δ224−274; Fig. 4b, c) exhibited deadenylation activity similar to those of fragments 1−248 and 102−313. In contrast, combining deletion of residues 249−274 to disrupt NOT and NOT9 binding (Δ249−274; Supplementary Fig. 6c, d) with CNBD loss markedly relieved IL-3 repression − an effect mirrored by the CNBD deletion plus the 3×W/A N-terminal mutations (Fig. 4b, d). Together, these results show that no single CCR4−NOT contact can sustain rapid deadenylation; rather, multivalent engagement of multiple modules is required for full TTP activity.

Thus, efficient and rapid TTP-mediated deadenylation requires multivalent engagement of either the TZF domain together with both N- and C-terminal IDRs (fragment 1−274; Fig. 3c) or the full C-terminal IDR alone (fragment 102−326; Fig. 4a).

Finally, to test multivalency in a cellular context, we over-expressed TTP variants in HEK-293T cells under a CMV promoter (Supplementary Fig. 6e) and measured three endogenous TTP targets[16] (IER3[48], CXCR4[49], and PIM3[50]) by RT-qPCR (Fig. 4e). Variants lacking two interaction interfaces (4×W/A + ΔCNBD and 1−223) showed a greater loss of repression than those missing only one site (ΔCNBD or 4×W/A), reinforcing that efficient mRNA decay depends on multivalent CCR4−NOT engagement by TTP.

## Phosphorylation does not affect TTP/ZFP36L1/2 binding to CCR4−NOT or deadenylation

Extensive phosphorylation of TTP and its family members ZFP36L1 and ZFP36L2 has been proposed to regulate their binding to AU-rich element (ARE)-containing mRNAs and recruitment of CCR-NOT[29,32−35,51,52] (Supplementary Fig. 7a). Prior work has focused on serines 60, 186, and 323 in human TTP (serines 52, 178, and 316 in mouse), showing that phosphorylation at these sites impairs TTP-dependent mRNA decay and weakens CCR4−NOT association[32−35].

We expressed TTP, ZFP36L1, and ZFP36L2 in HEK-293 cells and baculovirus-infected Sf21 insect cells to permit native phosphorylation, then compared these preparations, with and without phosphatase treatment, to bacterially produced, non-phosphorylated controls (Fig. 5a). Both mammalian- and insect-expressed proteins exhibited pronounced electrophoretic mobility shifts that were reversed by phosphatase, consistent with phosphorylation. Mass spectrometry further confirmed modification of TTP at the conserved Ser 60 and Ser

186 sites in both expression systems (Fig. 5b and Supplementary Data 1), although most potential phosphorylation sites within the C-terminal IDR were not analyzed due to gaps in our mass spectrometry coverage.

In vitro deadenylation assays revealed identical activities for phosphorylated versus dephosphorylated TTP (Fig. 5c), and phosphorylated ZFP36L1 and ZFP36L2 performed indistinguishably from TTP in promoting targeted deadenylation. These results indicate that phosphorylation does not appreciably alter TTP's RNA-binding activity under our conditions, nor does it disrupt direct engagement of any family member with the previously characterized CCR4−NOT modules (Fig. 5d). Both ZFP36L1 and ZFP36L2 maintained strong interactions with the extended NOT1:10:11 and NOT modules but failed to bind the NOT9 module in these assays, hinting at a TTP-specific CCR4−NOT binding profile.

Although phosphorylation did not directly alter CCR4−NOT-mediated deadenylation, we asked whether it might instead regulate TTP by recruiting phospho-serine/threonine−binding 14-3-3 proteins, which have been proposed to sequester TTP and inhibit TTP-targeted mRNA decay[29,32,53]. We first measured binding of phosphorylated TTP family members to 14-3-3β and 14-3-3η: all proteins interacted, but except for pZFP36L2 the associations were substoichiometric (Supplementary Fig. 7b). Strikingly, hyperphosphorylation of TTP by okadaic acid treatment in HEK-293T cells dramatically enhanced its binding to 14-3-3η (Supplementary Fig. 7c). We then tested whether 14-3-3 association impairs RNA binding, but hyperphosphorylated TTP alone failed to bind RNA (Supplementary Fig. 7d), consistent with prior observations[28], and thus precluding direct deadenylation assays under those conditions. In parallel, hyperphosphorylated TTP showed a marked loss of interaction with the extended NOT1:10:11 module (Supplementary Fig. 7e, f), likely due to CNBD serine phosphorylation[21,35]. Finally, because pZFP36L2 bound stoichiometrically to 14-3-3 even without phosphatase inhibition, we generated pZFP36L2:14-3-3 complexes and compared their RNA-binding and deadenylation activities to pZFP36L2 alone and observed no differences (Supplementary Fig. 7g, h). Together, these results indicate that, under both HEK-293T and Sf21 expression conditions, phosphorylation of TTP and its family members does not appreciably affect their CCR4−NOT interactions or their capacity to drive ARE-specific deadenylation.

## Phosphorylated TTP stimulates rapid shortening of PABPC1-coated poly(A) tails

Human TTP and its mouse ortholog can interact with PABPC1, as demonstrated by co-immunoprecipitation (co-IP) experiments[36,37]. This interaction was proposed to inhibit TTP-stimulated deadenylation[39] while being critical for mouse TTP-mediated translational repression[36].

We first tested PABPC1 binding using bacterially expressed TTP but observed no interaction, implicating a requirement for eukaryotic post-translational modification. Indeed, phosphorylated TTP purified from HEK-293T or Sf21 cells (pTTP) efficiently bound PABPC1, and this association was completely abolished by lambda phosphatase treatment (Fig. 6a). The same phosphorylation-dependence was seen for both pZFP36L1 and pZFP36L2 (Fig. 6b). Through deletion mapping, we pinpointed a discrete segment within the C-terminal IDR of TTP (residues 249−313) as necessary for PABPC1 engagement (Supplementary Fig. 8a−c), a region distinct from the mouse TTP homologous site previously reported[36].

By mapping PABPC1 truncations, we localized TTP binding to its RNA recognition motifs (RRMs), with the RRM2-3 fragment that uniquely contains both inter-RRM α-helical linkers essential for interaction with all human TTP family members (Fig. 6c−e; Supplementary Fig. 8d). Given the phosphorylation dependence, we hypothesized that surface-exposed basic residues in these linkers mediate

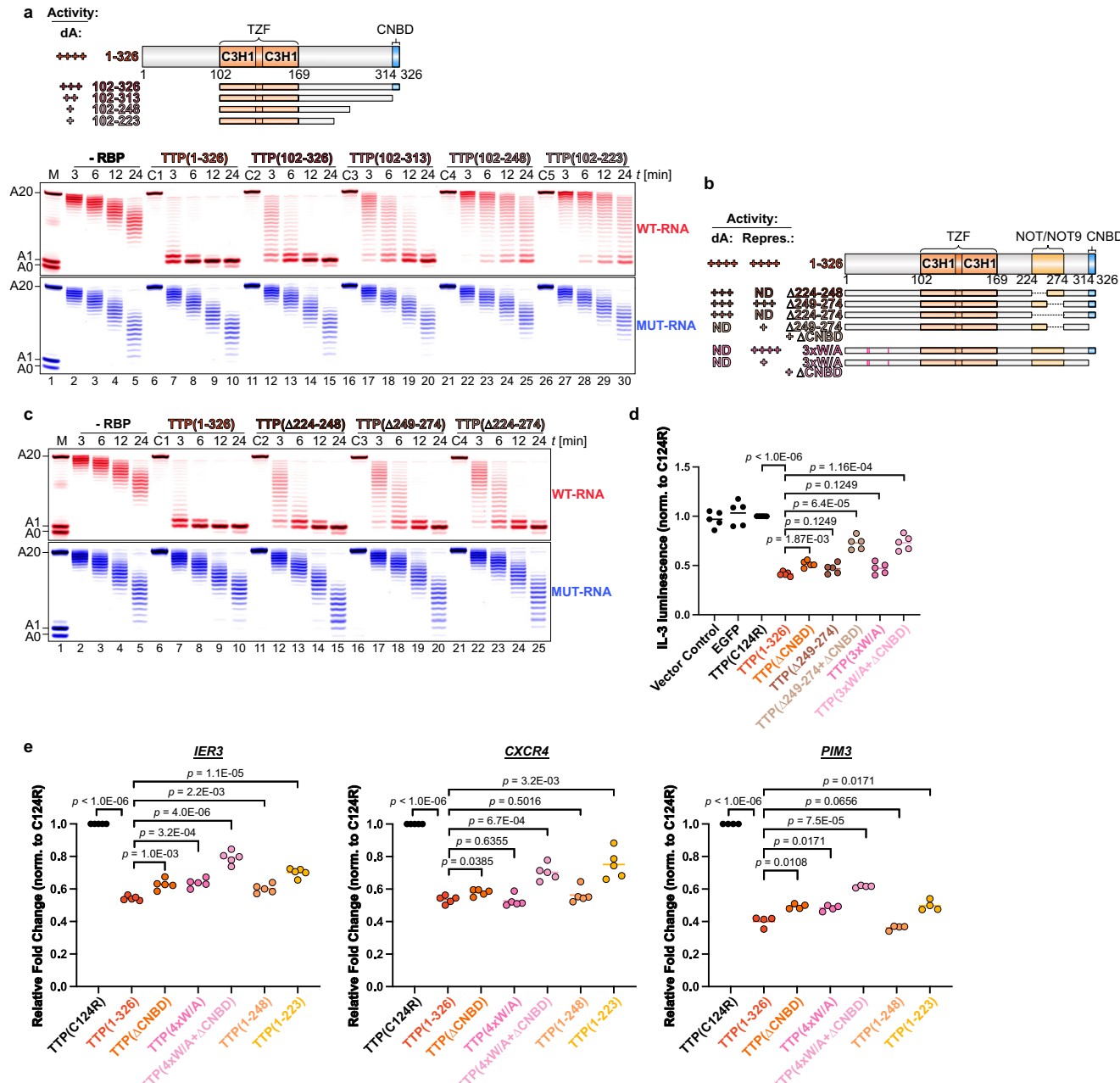

**Fig. 4 | Multivalent interactions between TTP and CCR4–NOT are crucial for rapid deadenylation and repression. a** Deadenylation assays assessing the ability of TTP fragments lacking both the N-terminal IDR and the indicated regions within the C-terminal IDR to stimulate targeted deadenylation. Domain organization schematic and associated deadenylation ('dA') activity summary is presented. All TTP proteins were added at 100 nM, in twofold excess over CCR4–NOT (50 nM) and substrate RNA (50 nM). Controls included reactions with RNA and the indicated TTP variant (C1–C5) alone. **b** Domain organization and summary of activity for TTP deletion and mutant variants used in deadenylation assays (**c**) and in co-transfection protein production assays (panel d). Mutated tryptophan residues (W32A/W38A/W69A) are indicated with pink bars. **c** Deadenylation assays testing deletion variants of the region in TTP that interacts with the NOT9 and NOT modules, assessing their ability to promote targeted deadenylation. The assay was performed as in (**a**). **d** The effect of deleting two critical CCR4–NOT-interacting regions in TTP on the repression of IL-3 protein production was measured via secreted IL-3 luminescence. Either both the CNBD and part of the C-terminal repressor region (Δ249–274) were deleted (light brown) or the CNBD was deleted in combination with point mutations of the three tryptophan residues (W32A/W38A/W69A) in TTP's N-terminal IDR (light pink). Each data point represents one biological replicate (*n* = 5). **e** RT-qPCR of endogenous *IER3*, *CXCR4*, and *PIM3* in HEK-293T cells transduced with CMV-driven vectors expressing various TTP variants: wild-type (1–326), RNA-binding mutant (C124R), ΔCNBD, all tryptophan residues to alanine mutant (4xW/A), a combination of ΔCNBD and 4xW/A, and C-terminal truncations (1–248 and 1–223). Each data point represents one biological replicate (*n* = 5 for *IER3* and *CXCR4*, *n* = 4 for *PIM3*). Statistical significance in (**d**, **e**) was determined using a two-tailed Student's *t*-test followed by Holm-Šidák correction for multiple comparison and shown as exact *p* values.

binding (Supplementary Fig. 8d, e), and indeed, reversing the charge in either the RRM2-3 or RRM3-4 linker nearly abolished TTP association; dual linker mutations eliminated it entirely (Supplementary Fig. 8f).

To test whether TTP and poly(A) compete for the same PABPC1 surface, similar to PAIP2A, which displaces PABPC1 from poly(A) tails to repress translation[54], we compared binding under two conditions: pre-formed poly(A):PABPC1 complexes and pre-formed TTP:PABPC1

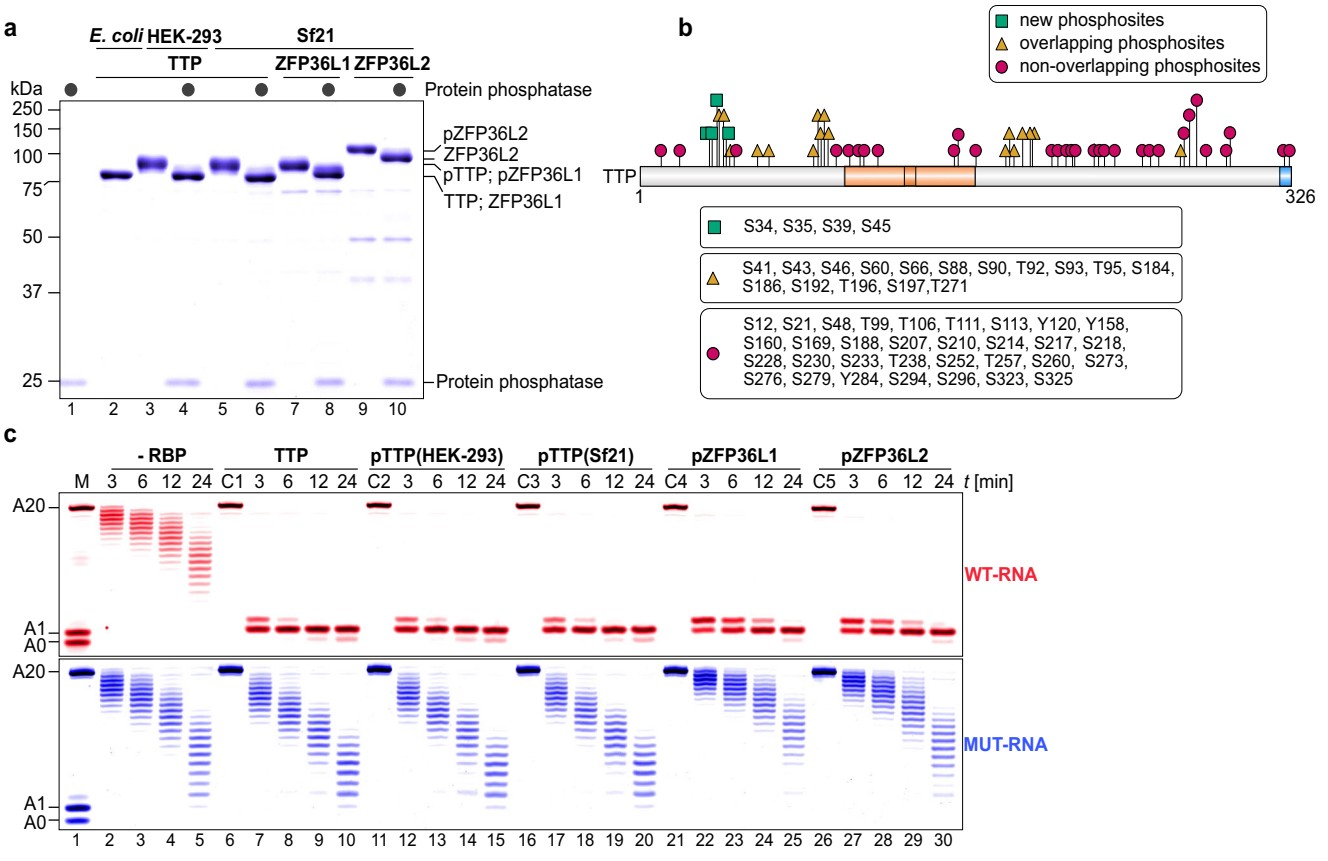

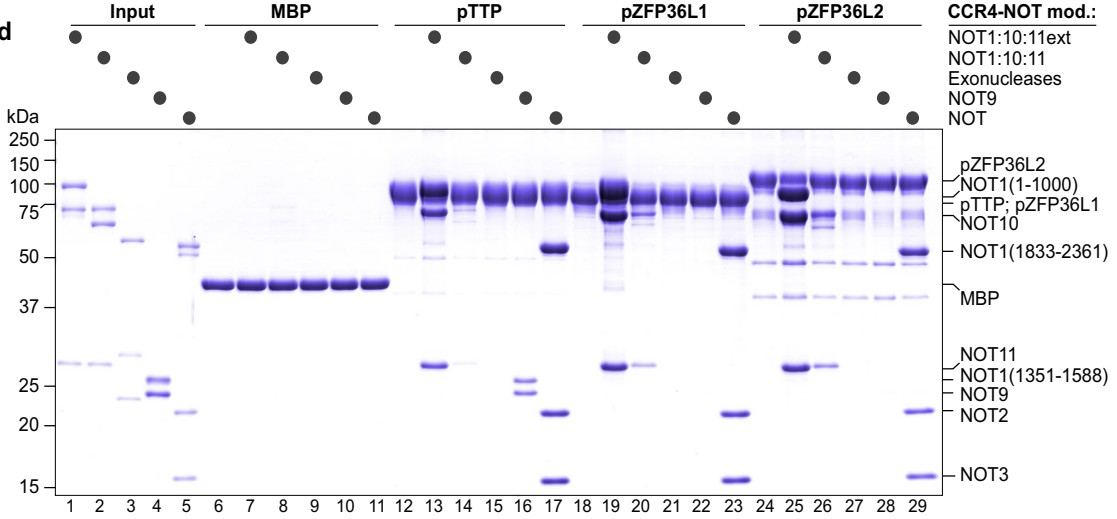

**Fig. 5 | Phosphorylation of TTP and its family members does not directly affect CCR4−NOT interaction or deadenylation activity. a** Phosphorylated TTP (pTTP) and its human family members (pZFP36L1, pZFP36L2) were expressed and isolated from eukaryotic systems (human HEK-293 and insect Sf21 cells). Phosphorylation status was confirmed by treating proteins with protein phosphatase (black circles) and comparing migration patterns to untreated samples via SDS-PAGE. TTP expressed in *E. coli* served as a non-phosphorylated control. **b** Diagram comparing previously reported phosphorylation sites in TTP based on the PhosphoSitePlus database[72] to sites identified by mass spectrometry in TTP isolated from HEK-293 or

Sf21 cells in this study. Overlapping sites (orange triangle), non-overlapping sites (red circles) and new sites (green rectangles) are indicated. **c** Deadenylation assays testing the activity of pTTP (expressed in HEK-293 or Sf21 cells) and phosphorylated pZFP36L1 and pZFP36L2 in targeted deadenylation. The activity of pTTP was also compared to non-phosphorylated TTP. All TTP proteins were used at 100 nM, in twofold excess over CCR4−NOT (50 nM) and substrate RNA (50 nM). Controls included reactions with TTP and RNA alone (C1−C5). **d** Pull-down assays demonstrating the interactions between phosphorylated TTP, pZFP36L1, and pZFP36L2, with CCR4−NOT modules (black circles). MBP was included as a negative control.

complexes. We found that TTP association was significantly reduced when PABPC1 was already bound to poly(A), whereas pre-assembled TTP:PABPC1 complexes remained stable even in the presence of excess poly(A) for at least one hour (Fig. 6f). These results demonstrate that

TTP and poly(A) engage PABPC1 in a mutually exclusive manner, consistent with competition for the same binding interface.

To evaluate how the phosphorylation-dependent TTP:PABPC1 interaction influences deadenylation, we performed ARE-specific

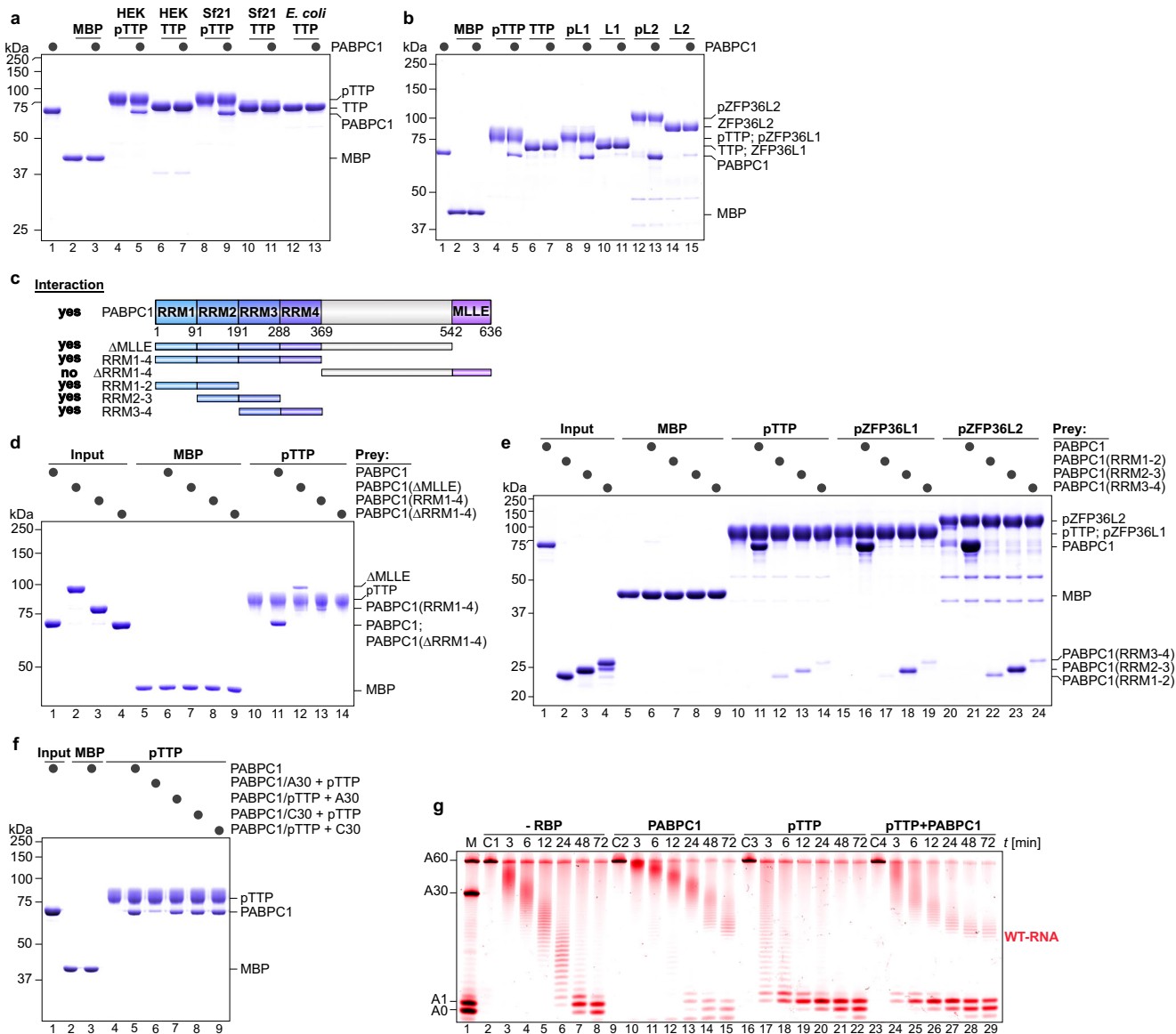

**Fig. 6 | Phosphorylation-dependent interaction between TTP and PABPC1 promotes CCR4–NOT-mediated shortening of PABPC1-coated poly(A) tails.** **a**, **b** Pull-down assays testing the interaction between the cytoplasmic poly(A)-binding protein (PABPC1) and either phosphorylated TTP (pTTP) expressed in HEK-293, Sf21, or *E. coli* cells, or phosphorylated human family members. Proteins were incubated with or without protein phosphatase (black circles) to assess the effect of dephosphorylation on the interaction. MBP was included as a negative control. **c** Domain architecture of PABPC1 and a summary of observed interactions between PABPC1 fragments and pTTP (**d**, **e**). RRM: RNA recognition motif; MLLE: mademoiselle protein–protein interaction domain. **d** Pull-down assays showing the interaction between pTTP and various fragments of PABPC1, as indicated (black circles). MBP was included as a negative control. **e** Characterization of the interaction between the RRMs of PABPC1 and pTTP, pZFP36L1, and pZFP36L2, via pull-down assays. **f** Assessment of PABPC's ability to interact simultaneously with a poly(A) tail and pTTP. Complexes of PABPC1 with a poly(A) tail of 30 adenosine residues (PABPC1/A30) were preformed before pTTP addition, or complexes of PABPC1 with pTTP (PABPC1/pTTP) were formed before adding A30 oligo. To confirm the poly(A) specificity, incubations were carried out with a C30 oligo. MBP was included as a negative control. **g** Deadenylation assay testing the effect of PABPC1 on targeted deadenylation by pTTP. A PABPC1:RNA complex was pre-formed first by addition of PABPC (150 nM) in threefold excess over substrate RNA (50 nM) to saturate the poly(A) tail. pTTP protein (100 nM) was added in twofold excess over CCR4–NOT (50 nM) and substrate RNA (50 nM). Controls included reactions lacking pTTP or PABPC1(-RBP), and reactions with only RNA and PABPC1 (C2) or pTTP (C3) or PABPC1 and pTTP (C4).

assays with CCR4–NOT, PABPC1, and phosphorylated TTP (pTTP) (Fig. 6g). In the presence of PABPC1 alone, deadenylation stalled at defined tail lengths A45 (Fig. 6g; lane 4–5 vs 11–12), A27 (Fig. 6g; lane 6–7 vs 13–14), and A15 (Fig. 6g; lane 7–8 vs 14–15), whereas inclusion of pTTP dramatically accelerated removal of these intermediates (lanes 11–15), consistent with pTTP displacing PABPC1 from poly(A). A small fraction of RNA remained resistant, stalling around A15 even after extended reaction times (lanes 27–29). Importantly, because these assays require low salt (50 mM) to maintain CCR4–NOT activity, we detected non-specific PABPC1:TTP associations irrespective of

phosphorylation status, which constrained our options to fully assess the role of the TTP:PABPC1 interaction in deadenylation under these conditions.

To assess how the TTP:PABPC1 interaction impacts mRNA stability in cells, we generated serine-to-alanine mutants targeting eight conserved phosphosites within the critical 249–313 segment of TTP's C-terminal IDR (Supplementary Fig. 9a, b). These S/A mutants lost both recombinant and endogenous PABPC1 binding yet retained full CCR4–NOT-stimulated deadenylation activity in vitro (Supplementary Fig. 9c). In HEK-293 cells, expression of the S/A variant produced a

modest derepression of IL-3 protein (Supplementary Fig. 9d) and a corresponding relief of mRNA repression for endogenous TTP targets *IER3*, *CXCR4*, and *PIM3* (Supplementary Fig. 9e). The enhanced deadenylation and decay of *Il3* mRNA by TTP and its family members in HEK-293 cells has been described previously, as has dependence of this enhanced decay on the presence of the AU-rich region in *Il3* mRNA[44]. We next asked whether 14-3-3 proteins could mask these phosphorylation sites in vivo, but excess 14-3-3η caused only a minor reduction in PABPC1 association for both pTTP and pZFP36L2 (Supplementary Fig. 9f).

Taken together, our data show that phosphorylation of specific serines in TTP's C-terminal IDR is required for productive engagement of PABPC1 and thereby lowers both mRNA stability and translational output. Importantly, these phosphorylation events do not measurably alter the efficiency of TTP-directed CCR4−NOT-mediated deadenylation itself. These results argue for a mechanism in which efficient ARE-specific deadenylation depends on TTP, both forming multivalent CCR4−NOT interfaces and displacing PABPC1 from the poly(A) tail.

## Discussion

We propose a mechanistic model (Fig. 7) in which tristetraprolin (TTP) leverages multivalent interactions with the CCR4−NOT deadenylase complex and the cytoplasmic poly(A)-binding protein (PABPC1) to ensure rapid post-transcriptional regulation of mRNA. In this model, TTP binds to target mRNAs through AU-rich elements (AREs) in their 3′ UTRs and simultaneously recruits CCR4-NOT via multiple interaction sites. This multivalent recruitment enhances the efficiency of deadenylation, as TTP can simultaneously engage CCR4-NOT via multiple sites and promote the shortening of the protective poly(A) tail, facilitated by its interaction with PABPC1. TTP helps to displace PABPC1 from the poly(A) tail, accelerating deadenylation and leading to rapid mRNA decay. The multivalent interactions illustrated in this model demonstrate the importance of TTP's ability to simultaneously interact with multiple partners to achieve robust and efficient mRNA destabilization, which elevates TTPs ability to quickly modulate gene expression in response to environmental cues or stress signals.

Although earlier studies identified TTP's conserved C-terminal NOT1-binding domain (CNBD) as key for CCR4−NOT recruitment[21,35], deleting this domain only modestly affected mRNA decay[22,41]. By identifying additional interaction sites within TTP's intrinsically disordered regions (IDRs), we provide a fuller understanding of how TTP engages CCR4−NOT. This multivalent binding likely enhances interaction stability and specificity, facilitating rapid and targeted deadenylation of ARE-containing mRNAs. Such multivalency may ensure robust mRNA regulation under varying conditions, allowing TTP to promote mRNA decay even when individual sites are compromised. This redundancy is crucial in dynamic cellular environments where proteins undergo modifications, interact with multiple partners, or compete for interaction sites. It seems almost certain that additional regulatory partners could modulate TTP's activity. Recently, a potential regulator named RESIST was discovered, presumably blocking TTP's association with CCR4−NOT during the type I interferon response[55]. Mutations or deletions of specific sites may lead to nuanced changes in gene expression, potentially contributing to disease states if regulatory balance is disrupted.

Although prior reports have argued that phosphorylation reduces TTP's CCR4−NOT binding and deadenylation activity[33–35], we find that phosphorylated TTP purified under standard mammalian or insect expression conditions retains full CCR4−NOT association and stimulates deadenylation. This discrepancy likely reflects differences in phosphorylation sites or assay conditions. Consistent with established roles for phospho-serine motifs, we confirmed robust, phosphorylation-dependent interactions between TTP (and ZFP36L2) and 14-3-3 proteins[32,53] in direct binding assays with recombinant proteins. Although 14-3-3 proteins were proposed to negatively regulate TTP-dependent decay of ARE-mRNAs[29,32], we observed unaltered deadenylation activity with preformed ZFP36L2:14-3-3 complexes. In contrast, phosphorylation is essential for TTP's engagement with PABPC1 and its displacement from the poly(A) tail, which could substantially influence mRNA repression. We speculate that differential phosphorylation of TTP may be a regulatory switch to facilitate the exchange of functional partnerships between CCR4−NOT recruitment and PABPC1 competition to enable precise, signal-dependent responsiveness of ARE-mRNA decay.

Similar multivalent interactions between TTP family members and CCR4−NOT complexes across species may provide evolutionary advantages by ensuring flexibility and resilience in mRNA regulation. The human TTP family members ZFP36L1 and ZFP36L2 also exhibited similar multivalent and phosphorylation-dependent binding, suggesting a conserved strategy. Additionally, multivalent interactions are common among RNA-binding proteins and regulatory complexes such as Pumilio[56,57], Roquin[58], and Unkempt[59]. Future studies of these mechanisms in various contexts could reveal universal principles of RBP-effector interactions that govern gene expression regulation.

In conclusion, our work reveals that TTP and its family members ZFP36L1 and ZFP36L2 exploit a network of multivalent, phosphorylation-regulated contacts to recruit the human CCR4−NOT complex and displace PABPC1, thereby driving selective and rapid deadenylation of RNAs bearing the UUAUUUAUU motif. By systematically mapping each protein's interface with CCR4−NOT modules and identifying the phosphorylation-dependent PABPC1 binding site, we uncover how combinatorial interactions fine-tune mRNA deadenylation and repression. We propose that this modular strategy — anchoring to target via the conserved tandem zinc-finger domain and amplifying repression via low-complexity regions and post-translational modifications — is a conserved feature of TTP family proteins,

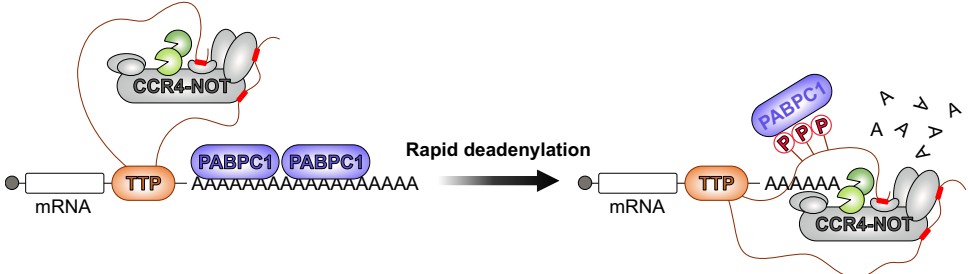

**Fig. 7 | A proposed model of TTP-mediated rapid deadenylation via multivalent interactions with CCR4−NOT and PABPC1.** Schematic model illustrating how TTP exploits multivalent interactions with two key regulatory effectors of mRNA fate — CCR4−NOT and PABPC1. Short segments within TTP's intrinsically disordered C-terminal region (red rectangles) serve as docking sites that scaffold these effectors and direct rapid, transcript-specific deadenylation. Phosphorylation of TTP (red "lollipop" symbols) promotes high-affinity binding to PABPC1, displacing PABPC1 from the poly(A) tail and thereby clearing access for CCR4−NOT to engage the tail and catalyze efficient deadenylation. Prepared in Affinity Designer 2.

enabling dynamic, sequence-specific control of gene expression across eukaryotes.

## Methods

### Plasmid constructs

Proteins and DNA constructs used in this study are listed and described in the supplementary material (Supplementary Tables 1,2). DNA constructs for the expression of CCR4–NOT complexes and modules were previously described[42,60–62]. The following section describes the generation of DNA constructs for the IL-3 HiBiT assay in more detail. All of the following constructs are under control of the CMV promoter.

**IL-3 HiBiT assay.** The pSK(CMV/mIL-3-HiBiT) construct was designed on the basis of the pSK-(CMV/mIL-3) construct, which was previously described[44]. A HiBit tag was inserted at the C-terminus of the mouse IL-3 protein (mIL-3) encoding the 11-residue HiBit peptide (VSGWRLFKKIS) immediately before the stop codon.

For the pBS + (CMV/sec-mCherry) construct, a double-stranded oligonucleotide encoding a 29-residue IL-6 signal peptide (MNSFSTSAFGPVAFSLGLLLVLPAAFPAP) was inserted into the Asp718 and SalI sites of the vector pBS + (CMV/BGH3′)[7], followed by a PCR fragment that encoded mCherry, in frame with the IL-6 signal peptide, at the SalI and NotI sites. The PCR template for mCherry, pCI/neo(CMV/MMP9-mCherry), was a generous gift from Jay Yang[63].

The vector construct pBS + (CMV/EGFP-BGH3′) was made by excising EGFP from pEGFP-NI (Clontech), using the restriction enzymes HindIII and NotI, and inserting the EGFP-encoding fragment into the same sites of vector pBS + (CMV/BGH3′). The construct pBS + (CMV/nes-EGFP-BGH3′) was made by inserting a double-stranded oligonucleotide encoding the 13-residue nuclear export sequence from human TTP into pBS + (CMV/EGFP-BGH3′), upstream and in-frame with EGFP.

**TTP mutant constructs for IL-3 HiBiT assay.** The pBS + (CMV/TTP-EGFP) construct was made by inserting the human TTP protein coding sequence (GenBank accession number NM_003407.5) into the Asp718 and AgeI sites in pBS + (CMV/EGFP-BGH3′), upstream and in-frame with EGFP. The C-terminal truncation mutants of human TTP (corresponding to residues 1–206, 1–248, 1–274, and 1–313) were made by replacing the C-terminus with shortened fragments at the internal native SmaI site (nt position 582 of NM_003407.5) and the vector AgeI restriction site that was in-frame with EGFP. These smaller fragments were produced by PCR, using a 5′-primer that contained human TTP sequences that encompassed the native SmaI restriction site and 3′-primers that included the TTP sequences with the intended truncation point followed by the AgeI restriction site.

The pBS + (CMV/TTP-EGFP) constructs with internal mutations in human TTP were made using the PCR primer-overlapping mutagenesis technique.

**Relocalization microscopy constructs.** The DNA constructs were adapted for use in mammalian cells from a previous study in insect cells[46]. For localization at the plasma membrane, we inserted the pleckstrin-homology (PH) domain of the rat phospholipase CD1 (PLCD1; NCBI RefSeq: NP_058731.2) with a C-terminal mCherry (GenBank: AAV52164.1) fusion into the mEGFP-C1 vector between the NheI and EcoRI restriction sites, thereby replacing the mEGFP (mEGFP-C1 was a gift from Michael Davidson (Addgene plasmid #54759)). To localize the fusion protein to the endoplasmic reticulum, we inserted the mini membrane protein subunit 4 of the yeast oligosaccharyl-transferase complex (OST4, NCBI RefSeq: NP_010049.1) with a C-terminal mCherry fusion into the mEGFP-C1 vector between the NheI and EcoRI restriction sites, replacing the mEGFP. Both of the generated constructs have a multiple cloning site downstream of the membrane anchor protein-mCherry fusion, allowing the insertion of other fusion proteins. We inserted human TTP between the EcoRI and BamHI sites of the pC1-PH-mCherry plasmid, downstream and in-frame with PH-mCherry. The first 117 residues of human NANOS3 (UniProtKB/Swiss-Prot: P60323.1) were inserted between the EcoRI and BamHI sites of the pC1-OST4-mCherry plasmid, downstream and in-frame with OST4-mCherry. NOT1(SHD), consisting of residues 1847 to 2361 of human NOT1 was inserted between the HindIII and BamHI restriction sites of the mEGFP-C1 vector, downstream and in-frame with mEGFP.

**TTP constructs for RT-qPCR.** For overexpression of human TTP and mutant variants in HEK-293T (DSMZ no. ACC 635) cells, we inserted the TTP variants with a C-terminal StrepII-tag between the NheI and BamHI restriction sites of the mEGFP-C1 vector, replacing the mEGFP.

**Recombinant protein production and purification.** CCR4–NOT complexes and modules were expressed in either bacteria or insect cells, purified by affinity chromatography, and reconstituted, followed by either ion exchange or size exclusion chromatography[42,58,62].

TTP constructs were expressed in *E. coli* BL21(DE3) Star cells (Thermo Fisher Scientific) in autoinduction media[64] at 20 °C overnight as fusion proteins carrying an N-terminal, TEV-cleavable MBP-tag, and a C-terminal StrepII-tag. Harvested cells were resuspended in protein buffer (50 mM HEPES pH 7.5, 300 mM NaCl, 10% sucrose), and lysed by sonication. The lysate was clarified by centrifugation at 40,000 $g$ for 40 min and loaded on a 1 ml streptavidin-charged StrepTrap XT column (Cytiva). Contaminants were removed by washing with high salt buffer (50 mM HEPES pH 7.5, 1000 mM NaCl, 10% sucrose) before elution with lysis buffer supplemented with 50 mM biotin. Eluted protein was further purified by size exclusion chromatography on a Superdex 200 26/600 column (Cytiva) in protein buffer supplemented with 2 mM DTT. The peak fractions were then pooled together, concentrated with a centrifugal filter, flash-frozen in liquid nitrogen, and stored at −80 °C.

Phosphorylated TTP constructs were expressed either in Sf21 (Thermo Fisher Scientific catalog no. 11497013) or HEK-293 (ATCC catalog no. CRL-1573) cells. For Sf21 insect cells, the MultiBac baculovirus expression system[65,66] was utilized as previously described[42]. In short, Sf21 cells were grown to a density of $2 \times 10^6$ cells/ml at 27 °C in Sf900II medium (Thermo Fisher Scientific), infected with the V1 TTP variant stock of baculovirus, and harvested 48 h after they stopped dividing. HEK-293 cells were cultured in MEM supplemented with 10% fetal bovine serum (FBS), 2 mM L-glutamine, and 100 U/ml penicillin-streptomycin in a humidified incubator at 37 °C with 5% CO$_2$ and atmospheric oxygen and transiently transfected using the calcium phosphate method as described previously[67]. Briefly, cells were seeded in 100 mm culture dishes with $7.5 \times 10^5$ cells per dish one day before the transfection. To each dish, 150 ng of pBS + (CMV/MBP-TTP-Strep) DNA was added together with 4,850 ng of vector pBS+ DNA as calcium-phosphate precipitates in 1 ml transfection solution (140 mM NaCl, 25 mM HEPES pH 7, 0.12 mM CaCl$_2$, and 0.75 mM sodium phosphate. 24 h after adding the DNA, the transfection medium was replaced with fresh culture medium and incubated for an additional 24 h before treating the cells with PMA. PMA (final concentration 100 nM) was added to the treatment group for 3 h, and the control cells were incubated with equal volumes of DMSO. The cells were harvested and lysed in HEK-293 lysis buffer (10 mM HEPES pH 7.5, 40 mM NaCl, 12 mM NaF, 0.2% N-P40, 5% sucrose with freshly added 0.1 mM PMSF, 1 mg/ml Pepstatin A, and 8 mg/ml leupeptin). The lysates were centrifuged at 13,000 $g$ for 15 min. The buffer composition of the clarified extract was adjusted to 7.8 mM HEPES pH 7.5, 400 mM NaCl, 9.4 mM NaF, 2 mM MgCl$_2$, 25 μM ZnSO$_4$, and 10% sucrose. The protein purification of pTTP was performed as described above.

All PABPC1 constructs were expressed in *E. coli* Bl21(DE3) Star cells (Thermo Fisher Scientific) in autoinduction media[64] at 20 °C overnight as fusion proteins carrying an N-terminal His-tag. Harvested cells were

resuspended in lysis buffer (50 mM HEPES pH 7.5, 1000 mM NaCl, 5% glycerol, 25 mM imidazole), and lysed by sonication. The lysate was clarified by centrifugation at 40,000 $g$ for 40 min and loaded on a 5 ml nickel-charged HisTrap column (Cytiva). Contaminants were removed by washing with lysis buffer supplemented with 40 mM imidazole, and PABPC1 constructs were eluted in lysis buffer supplemented with 250 mM imidazole. PABPC1 constructs were further purified by size exclusion chromatography on a Superdex 75 or Superdex 200 26/600 column (Cytiva) in a buffer containing 10 mM HEPES pH 7.5, 200 mM NaCl, 5% glycerol, 2 mM DTT. Peak fractions were then pooled together, concentrated with a centrifugal filter, flash-frozen in liquid nitrogen, and stored at −80 °C.

Both 14-3-3 isoforms were expressed and purified as described above for the PABPC1 constructs, except that the lysis buffer contained 300 mM NaCl instead of 1000 mM.

### Pull-down assays
Purified proteins were immobilized as bait either via their C-terminal StrepII-tag on homemade streptavidin resin or in the case of pull-downs with the NOT1:10:11 extended module via their N-terminal MBP-tag on amylose resin (NEB). 250 pmol of bait protein was incubated for 1 h in pull-down buffer (50 mM HEPES pH 7.5, 200 mM NaCl, 0.03% Tween-20) at 6 °C under constant agitation. Unbound protein was removed following two washes with pull-down buffer, and 500 pmol of prey protein was incubated for 1 h with the bead-bound protein. Finally, the beads were washed three times with binding buffer, and the bound proteins were eluted using binding buffer supplemented with either 50 mM biotin or 30 mM D-(+)-maltose. Eluted proteins were analyzed using SDS-PAGE, followed by Coomassie blue staining.

The competition experiment (Fig. 6f) between pTTP, PABPC1, and A30 was performed as follows. First, PABPC1 (250 pmol) was incubated either with pTTP (500 pmol) and homemade streptavidin beads or with a poly(A) tail of 30 adenosine residues (500 pmol) for 1 h in pull-down buffer at 6 °C. In parallel, in a separate reaction pTTP (500 pmol) was immobilized on homemade streptavidin resin for 1 h as described above. The preformed PABPC1:A30 complex was added to the bead-bound pTTP, while the A30 oligo (500 pmol) was added to the pre-formed PABPC1:pTTP complex. All reactions were incubated for an additional hour and further processed as described above. To confirm the poly(A) specificity, incubations were carried out with a C30 oligo instead of the A30 oligo. In all pull-down assays, His$_6$-MBP-StrepII was immobilized on beads as a negative control.

For pull-down experiments with transfected HEK-293T cells, ca. 2 × 10$^6$ cells were lysed in pull-down buffer supplemented with 1 mM PMSF. TTP proteins were carrying an N-terminal EGFP-MBP-tag and were immobilized via their C-terminal StrepII-tag on homemade streptavidin resin for 2 h in the presence of RNase A (150 μg/ml) at 6 °C under constant agitation. The beads were washed six times to remove unbound protein. Bound protein was eluted with pull-down buffer supplemented with 50 mM biotin. Eluted proteins were analyzed by western blot.

### Deadenylation assays
Deadenylation assays were performed as described before[42,68] with minor modifications. Briefly, reactions were carried out in 40 μl at 37 °C in a reaction buffer containing 20 mM PIPES pH 6.8, 10 mM KCl, 40 mM NaCl, and 2 mM Mg(OAc)$_2$. Both 50 nM of RNA with the wildtype ARE (ARE-WT-RNA) carrying an Atto 532 label at the 5′ end and 50 nM of the mutated ARE (ARE-MUT-RNA) carrying a 6-FAM label at the 5′ end were mixed (sequences are listed in Supplementary Table 3). The substrate RNAs were either directly incubated with 50 nM CCR4−NOT (250 nM in the case of the NOT6:NOT7 heterodimer) or pre-incubated with 100 nM of the indicated TTP variant for 15 min before deadenylation was started by the addition of deadenylase complex. The reactions were stopped at the indicated time points by adding three times the reaction volume of

RNA loading buffer (95% deionized formamide, 17.5 mM EDTA pH 8.0, 0.01% bromophenol blue) and kept on ice until further analysis. In experiments with PABPC1, a single RNA containing the wild-type ARE with a poly(A) tail of 60 adenosine residues was used. PABPC1 (150 nM) was incubated for 15 min with 50 nM substrate RNA prior to the addition of 150 nM TTP. After a 15 min incubation, deadenylation reactions were initiated upon adding 50 nM CCR4−NOT.

The reaction products were separated on denaturing Tris-Borate-EDTA(TBE)-urea polyacrylamide gels (20% 19:1 acrylamide-bis acrylamide, 7 M urea, 1xTBE buffer) at 300-350 V for ca. 2 h. Fluorescent reaction products were visualized by scanning with a Typhoon RGB biomolecular imager (Cytiva).

All deadenylation experiments were performed in triplicate with representative gels shown. All assays were extensively validated with multiple batches of purified proteins.

### AlphaFold-Multimer predictions
Predictions were generated with AlphaFold-Multimer[69,70] version 2.3.2 with these key settings:

--db_preset=full_dbs  --max_template_date=2020-05-14  --models_to_relax=best  --model_preset=multimer  --num_multimer_predictions_per_model=5

The resulting predicted models were aligned in ChimeraX v1.8 to assess prediction convergence, and this software was used to prepare all structural figures.

### Dephosphorylation of TTP proteins
250 pmol of the phosphorylated versions of TTP, ZFP36L1, or ZFP36L2 (pTTP, pZFP36L1, or pZFP36L2) prepared from HEK-293 or insect cells were incubated with 800 units of lambda protein phosphatase (NEB) in 1x PPase buffer (NEB) and 1 mM MnCl$_2$ at 30 °C for 30 min. If not used further in pull-down assays, the reactions were stopped by the addition of an equal volume of 2x SDS-loading buffer (4% SDS, 20% glycerol, 10% β-mercaptoethanol, 0.02% bromophenol blue, 0.25 M Tris HCl pH 6.8).

### Maintenance and transfection of HEK-293 cells
HEK-293 cells were maintained and transiently transfected using the calcium phosphate method as described (see above), with the modifications described below.

One day before the transfection, HEK-293 cells were seeded at 1.25 × 10$^5$ cells/well into 6-well plates in 3 ml per well of 10% FBS/MEM. On the day of transfection, each transfection suspension was prepared as follows: The total DNA to be added to each well was 675 ng, consisting of 20 ng of pSK(CMV/mIL-3-HiBit) and 1.72 ng of pBS + (CMV/sec-mCherry) to monitor transfection efficiency, with the total DNA content being made up with the plasmid Bluescribe (pBS + ). For conditions in which the effects of TTP or its mutants were to be tested, 0.2 ng/well of pBS + (CMV/TTP-EGFP) or the various human TTP mutants were used. Since TTP-EGFP expressed in this system and its mutants are predominantly cytosolic, the construct pBS + (CMV/nes-EGFP) (0.2 ng/well) was included in three wells in each experiment to express a cytosolic form of EGFP to serve as a negative control for TTP. Three wells were also transfected with 675 ng/well of vector pBS+ alone as additional negative controls. For each transfection condition, a DNA set for four wells was prepared in 1 ml of calcium phosphate precipitate, and 0.25 ml of the suspension was added to each of the three wells.

24 h after the transfection, the transfection medium was removed and replaced by 5% FBS in Opti-MEM without phenol red (Invitrogen 11058021) for 1.5 ml/well and incubated for an additional 24 h. The medium was then again changed to 1.5 ml/well of fresh 5% FBS/Opti-MEM and cells were incubated overnight before harvesting of culture supernatant.

For transfection of HEK-293T cells for RT-qPCR and pull-down experiments, cells were seeded at 0.5 × 10$^6$ cells/well into 6-well plates in 2 ml per well of 10% FBS/DMEM. After 24 h, cells were transfected

with a mixture consisting of 10 µg plasmid DNA and 20 µl Lipofectamine 2000 (Invitrogen) in 200 µl Opti-MEM I supplemented with GlutaMAX (Gibco). Cells were harvested 24 h post-transfection.

Treatment with okadaic acid (PubChem CID: 446512, CAS Registry no. 78111-17-8, purchased from Enzo) was done at 1 µM for 90 min in 10% FBS/DMEM. Okadaic acid is a potent inhibitor of serine/threonine protein phosphatases PP2A ($IC_{50} \approx 0.1$ nM) and PP1 ($IC_{50} \approx 15$–20 nM), but it does not appreciably inhibit tyrosine, acid, or alkaline phosphatases. It is widely used as a chemical probe of PP1/PP2A function, but lacks the >30-fold selectivity over related phosphatases required for a "high-quality" probe definition as per adopted community criteria[71].

### IL-3 HiBit assay

Secreted mCherry and HiBit-tagged IL-3 were measured 16 h after the last media exchange. To do this, 500 µl of culture supernatant was removed from each well and centrifuged for 10 min at 13,000 x *g* to precipitate cells and debris. From this supernatant, 100 µl was added to a well of a half-area black 96-well plate (Corning 3993) to measure mCherry fluorescence (excitation at 560 nm and emission at 620 nm, with a 20 msec integration time and a 1 sec settle time) in a microplate reader (Tecan Infinite 200Pro). The average fluorescence measurement from the three wells transfected with vector pBS+ alone was usually 6 to 8% of the fluorescence measured from the wells where various DNA plasmids had been transfected. Therefore, the average measurement from the pBS+ alone transfected samples was subtracted from the values from the wells.

To quantify the secreted HiBit-tagged IL-3, the Nano-Glo HiBit Extracellular Detection System (Promega) was used. Cleared culture supernatant (25 µl) together with 50 µl of PBS were added to wells of black 96-well plates (Thermo Scientific 137101). An aliquot of 75 µl Nano-Glo mixture (Promega N2421), consisting of 75 µl of Nano-Glo HiBit Extracellular Buffer and the LgBit Protein (1:100) and the Nano-Glo HiBit Extracellular substrate (1:50), was added to each well. The samples were then placed on an orbital shaker for 10 min at 200 rpm, followed by measuring luminescence in the Tecan microplate reader with automatic attenuation time and 100 msec of integration time. The average measurement from the three wells transfected with vector pBS + alone was negligible and was usually 0.02 to 0.05% of the luminescence measured from the wells to which various active DNA plasmids had been transfected; this was therefore not subtracted from the luminescence readings.

Experiments were performed as five biologically independent repeats. Statistical significance was evaluated by two-tailed Student's *t*-test using GraphPad Prism v10.3.1. P values were corrected for multiple comparisons using the Holm-Šidák method.

### Real-time quantitative PCR

Total RNA was isolated from transfected HEK-293T cells 24 h post-transfection using the RNeasy kit (Qiagen) followed by an on-column DNase digestion. cDNA was synthesized using the iScript Supermix Kit (Bio-Rad) with 1000 ng RNA per reaction. 2 µl of 5-fold diluted cDNA was mixed with 0.5 µM qPCR primers in 20 µl of Power SYBR Green PCR master mix (Applied Biosystems) and amplified with a CFX Opus 96 (Bio-Rad). The relative fold changes were calculated using the ΔΔCt method with Rplp0 as the reference gene. qPCR primer sequences are listed in Supplementary Table 4.

Experiments were performed as five (*IER3* and *CXCR4*) or four (*PIM3*) biologically independent repeats. Statistical significance was evaluated by two-tailed Student's *t*-test using GraphPad Prism v10.3.1. *P* values were corrected for multiple comparisons using the Holm-Šidák method.

### Statistics and reproducibility

If not otherwise stated, representative data from at least three independent experiments (biological and technical replicates) is shown.

### Resource availability

Further information and requests for resources and reagents should be directed to and will be fulfilled by E.V., subject to a completed materials transfer agreement.

### Reporting summary

Further information on research design is available in the Nature Portfolio Reporting Summary linked to this article.

## Data availability

All data supporting the findings of this study are available within the paper and its Supplementary Information. The raw data associated with the gel images, deadenylation assays, cell-based assays can be found in the Source Data file. Specific data P values and statistical analysis are also included within the Source Data file. The raw and processed mass spectrometry proteomics data have been deposited to the MassIVE repository under accession MSV000098386 [https://doi.org/10.25345/C5FQ9QJ40]. The raw microscopy image data generated in this study has been deposited in the figshare database on https://doi.org/10.6084/m9.figshare.29451836. Source data are provided with this paper.

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

## Acknowledgements

We thank Anna Valkov and the Protein Technology Core staff for providing reagents and advice to support this work. We thank Mandy Jeske for providing the PH-mCherry and OST4-mCherry template plasmids for cloning of the relocalization assay constructs. We thank Stephen Lockett and the CCR Optical Microscopy and Analysis Lab (OMAL) for support with microscopy. We thank Ronald Holewinski and Thorkell Andresson at the NCI-Frederick Protein Characterization Laboratory for the mass spectrometry analysis. We also thank Sergey Tarasov and Marzena Dyba for their assistance with biophysical measurements. We thank Yevgen Levdansky for assisting J.C. with the initial experiments. We are also grateful to our colleagues at the RNA Biology Laboratory for their support and advice, and especially members of Colin Wu's laboratory for assistance with qPCR. This research was supported by the Intramural Research Program of the National Institutes of Health (NIH). The contributions of the NIH authors were made as part of their official duties as NIH federal employees, are in compliance with agency policy requirements, and are considered Works of the United States Government. However, the findings and conclusions presented in this paper are those of the authors and do not necessarily reflect the views of the NIH or the U.S. Department of Health and Human Services. F.P. was supported by a Walter Benjamin postdoctoral fellowship from the German Research Foundation (Deutsche Forschungsgemeinschaft) [Project number 531520533].

## Author contributions

Conceptualization: F.P., E.V. Methodology: F.P., W.S.L. Investigation: F.P., W.S.L., J.C., and K.L. Formal analysis: F.P., W.S.L. Visualization: F.P. Project administration: E.V., P.J.B. Resources: S.N.H. Supervision: E.V., P.J.B. Writing – original draft: F.P., E.V., P.J.B. Writing – review & editing: F.P., E.V., P.J.B.

## Competing interests

The authors declare no competing interests.
