## [Transparent Peer Review file · Nature Communications]

Multivalent Interactions with CCR4-NOT and PABPC1 Determine mRNA Repression Efficiency by Tristetraprolin

Corresponding Author: Dr Eugene Valkov

Version 0:

Reviewer comments:

Reviewer #1

(Remarks to the Author)

The authors of this manuscript focus on analyzing different regions of the human protein tristetraprolin (TTP) that are important for interacting with CCR4-NOT and for eliciting degradation of a labile AU-rich RNA. The authors report two main discoveries. One complements earlier reports that additional regions of TTP are implicated in binding CCR4-NOT; the other is that PABPC1 associates with TTP in a manner dependent on TTP phosphorylation, and this interaction removes PABPC1 from the poly(A) tail, further contributing to RNA decay. Through these studies, the authors propose additional details of TTP-elicited degradation of AU-rich mRNAs.

Given the great importance of understanding the function of RNA-binding proteins through their association with other proteins and post-translational modification, the topic addressed in the current manuscript is timely and significant. Unfortunately, over a dozen articles have reported functional interactions between TTP and CCR4-NOT for the past decade, which reduces the novelty of the authors' findings, although they do uncover additional structural details of the interaction of these proteins. In the last figure, the authors introduce a new protein interacting with TTP physically and functionally, PABPC1; although this finding is certainly exciting and novel, it is far too preliminary. One final concern is that the entire piece is based on biochemical *in vitro* analysis of recombinant TTP, CCR4-NOT, and PABPC2, and a synthetic labile RNA. To fully support their model, and the impact of the newly identified interactions, the authors need to test endogenous proteins and mRNAs. These concerns are explained below.

Main concerns

1. Figure 1: if the authors wish to connect the functional consequences of the 13 C-terminal residues of TTP and the tryptophan mutations in cells, to the effect of CCR4-NOT on the stability of IL-3 mRNA, they should analyze IL-3 mRNA levels and half-life, not only the levels of secreted IL-3.
2. The authors rely extensively on *in vitro* deadenylation assays, which are quick and convenient. However, whether these assays reflect *in vivo* deadenylation and decay of specific mRNAs in this paradigm, as regulated by these proteins, needs to be tested too. Studying how these interactions affect the stability of TTP targets (at a minimum IL-3 mRNA) in cells is essential to complement the data in Figures 1, 3 and 4.
3. The data in Figure 6 are the most novel and potentially exciting, but they open many questions for the model. Does PABPC1 binding cooperate with CCR4-NOT binding to a target mRNA? do these binding activities synergize? or perhaps compete? What specific TTP sites are phosphorylated in this paradigm of displacing PABPC1 from poly(A) tails. Furthermore, the interactions invoking PABPC1, physical and functional, need to be studied *in vivo* in cells, as well as on cellular mRNAs, not only using *in vitro* binding of recombinant proteins and decay assays.

Minor comments

1. The block letters used in some of the figures (black outline, orange fill) would be more visible if displayed in a solid color.
2. The title, why do the authors use the word 'Repression' instead of the actual process they measure, which is 'Decay'? in fact, they do not actually measure any mRNA, only partial RNAs.

Reviewer #2

(Remarks to the Author)

The manuscript by Pekovic et al investigates the TTP mediated RNA decay via its interactions with the CCR4-NOT complex and PABPC1. The authors present a body of work which reveals interesting novel findings with regards to the molecular mechanism of RNA decay mediated by TTP. The authors characterize the low complexity N- and especially C-terminal regions of TTP to be crucial for recruitment of RNA to the NOT complex. In contrast they find that the CCR4-NOT targeting domain of ZFP36 seems to be dispensable for CCR4-NOT mediated RNA deadenylation.

Also, the authors find that phosphorylation of TTP does not affect its RNA decay promoting function which is in contrast to previous models described for the role of TTP phosphorylation. Phosphorylation is however important for TTP interaction with PABPC1 which results in destabilization of RNA. This is an important finding.

The data presented here is interesting and challenges existing models of the molecular mechanisms of TTP mediated RNA decay. However, the manuscript could offer more to demonstrate that the mechanism is physiologically relevant.

Specific comments

The authors identify NOT2 and NOT3 as components of CCR4-NOT complex to interact with N terminal low complexity regions of TTP. Multiple proteins are associating with the CCR4-NOT complex with potentially even higher affinities than the interaction described here. An experiment in a cellular system which demonstrates this interaction in the presence of multiple different competing proteins would strengthen the author's case.

The authors show that TTP phosphorylation does not inhibit its ability to promote RNA deadenylation. It is known that TTP can be phosphorylated on multiple sites, some sites have been studied in cells and reported to promote TTP stability and to inhibit its ability to promote mRNA decay. How do the phosphorylation events induced in this work relate to the known phosphorylation sites? Which sites are phosphorylated? Can this explain the discrepancy with published work investigating individual phosphorylation sites and the data shown here which seems to induce the phosphorylation of all sites? In a more complex, live cell setting, where only some sites are phosphorylated, this may become relevant.

The authors should, at least, discuss that phosphorylated TTP can be bound by 14-3-3 proteins which can influence TTP function. Could addition of a 14-3-3 protein prevent phosphorylated TTP from promoting deadenylation? Might 14-3-3 protein block the TTP interaction with PABPC1?

The authors describe the interaction of phosphorylated TTP with PABPC1 as displacing PABPC1 to promote deadenylation. Identifying the relevant phosphorylation sites that mediate this interaction would add value to the study. In Figure 1F the authors use a HEK-293 transduction system to investigate the role of modified TTPs on the production of IL3. It is unclear whether altered IL3 production is due to changes in mRNA deadenylation. Also, it is worth clarifying whether the HEK-293 cells express any of the TTP family proteins which might also affect the assays.

Minor comments:

Line 125 please clarify what the components of the fully reconstituted human CCR4NOT complex include, as it is unclear what its composition is with respect to NOT4 from Ref 51.

Line 159 the NOT module referred to appears to consist of NOT2 and NOT3 and also the HEAT domain of NOT1? Please clarify here as in the paragraph multiple HEAT domains are referred to.

Line 180 it is not clear how the authors estimate significance when they claim that the TZF domain significantly enhanced targeted deadenylation activity.

Does the TZF domain of TTP used here in these experiments actually bind to RNA? Or does it bind RNA but just is not recruiting it to the CCR4-NOT complex? This is not completely clear.

Line 207 please indicate that the full length TTP is 1-326 for more clarity.

Line 209 it is not clear what the authors are referring to. Is it fragment 1-233? Please refer to the fragments as described in the figure. If it is fragment 1-233, it seems to me that the conclusion of the experiment is not correct.

Line 259 Can the authors please clarify whether they mean phosphatase inhibitors when they write "...phosphatase-treated purified proteins to bacterially expressed non phosphorylated controls".

Line 265 the authors state that phosphorylation did not affect TTPs RNA binding. However actual binding to RNA by TTP is not shown as far as I can tell.

Line 310-312, clarify that the deadenylation assays contain the CNOT complex as the "reactions containing only PABPC1" was slightly confusing.

Figure Legends 1G and 3E and 4D. Please clarify what each dot represents.

Please also state in all figure legends how many times each representative experiment has been performed.

Please also indicate what each figure panel represents. If Fig.1B shows a Coomassie gel, please indicate that. Same for Panels like 1C.

Please explain the rationale for why in some experiments HEK-293 cells are used and in other experiments Sf21 insect cells are used.

Reviewer #3

(Remarks to the Author)

Reviewer #4

(Remarks to the Author)

In this manuscript, authors investigated how tristetraprolin (TTP) interacts with CCR4-NOT and PABPC1 to control mRNA stability. Using purified proteins and in vitro assays, authors discovered that TTP engages CCR4-NOT through multiple interaction sites and promotes its deadenylation activity on mRNAs containing AU-rich elements. Authors also observed that phosphorylation of TTP did not affect its interaction with CCR4-NOT or its deadenylation activity, but was essential for TTP's binding to PABPC1.

This manuscript is an interesting piece of work. It is well-written and logically presented. However, data does not seem to fully support the authors' main conclusions. Some issues should be clearly addressed before publication.

Specific comments:

Regarding the model proposed, authors say "TTP promotes the processive deadenylation activity of CCR4-NOT on mRNAs containing AU-rich elements, with phosphorylation-dependent interactions with PABPC1 potentially enhancing deadenylation and promoting regulated mRNA decay". More experimental results should be included to confirm that interactions of phosphorylated TTP with PABPC1 enhance deadenylation and promote regulated mRNA decay.

Also, it would be interesting to show how the proposed mechanism of controlling mRNA decay responds to inflammation in the cell. In fact, the manuscript does not present in cellulo experiments and this would reinforce conclusions about the cellular role of TTP in the control of mRNA decay in response to cellular stimuli, such as inflammation.

Minor comment:

Authors should specify the type of mRNA repression.

Reviewer #5

(Remarks to the Author)

NCOMMS-24-72360-T

Pekovic et al investigate determinants required for interactions between CCR4-NOT, PABPC1 and TTP in a manuscript entitled 'Multivalent Interactions with CCR4-NOT and PABPC1 Determine mRNA Repression Efficiency by Tristetraprolin'. In this work, they primarily rely on reconstitution biochemistry, alphafold predictions, and RNA decay assays to investigate multivalent interactions between TTP and CCR4-NOT which serves to recruit deadenylase activity to model RNA substrates. Of note, phosphorylation of TTP was thought to promote recruitment of CCR4-NOT to enhance deadenylation, but the authors found that phosphorylation had no effect on interactions with CCR4-NOT. Instead, phosphorylation appears to enhance TTP association with the polyA binding protein PABPC1, an unexpected but very nice result which explains how phosphorylation may promote deadenylation.

The biochemical assays are well presented, especially with respect to analysis of RNA mixtures (containing mutations in recognition motifs) as the authors compared rates of decay in assays with WT and mutant uniquely labeled to enable analysis of the control decay substrate in the same reaction.

This is a clean and concise study that illuminates a complex network of interactions that underlie recruitment of factors to elicit deadenylation and promote decay. There are no major concerns with this manuscript in its current form, and only a few minor suggestions are offered.

In the abstract, the authors state: 'Here, we investigated how TTP interacts with CCR4-NOT and PABPC1 to control mRNA stability.' Given that most of the work is done on synthetic RNAs and not on mRNA per se, the authors might restate 'control mRNA stability' to 'promote RNA deadenylation'.

One line 248 the sentence starts with 'This study addressed...' and they then reference 7 references at the end of the sentence. What does 'This study' refer to?

The authors use the terms 'completely abolished' on lines 286 and 303. Abolished is an absolute term, it should not be prefaced with completely or any other qualifier.

Consider removing the eyeballs on CNOT6/7 in Fig 2a, maybe ok for a review, but seems a bit too cute for a primary

research manuscript.

Consider adding schematic/cartoon indicating a model/role for phosphorylation in the context of these interactions.

Version 1:

Reviewer comments:

Reviewer #1

(Remarks to the Author)

I appreciate the comprehensive responses from the authors. I have no further concerns.

Reviewer #2

(Remarks to the Author)

The authors provide a comprehensive revision which addresses the points raised. Overall, this is a significant body of work which shows how phosphorylated TTP can displace PABP and promote deadenylation by CNOT. There is still some scope to offer a bit more clarity. This could be to place the work in the context of other phosphorylation events which are outside of 249-313, especially those 60 and 186. Related to this is the observation that chemical induction of hyperphosphorylation of TTP results in loss of RNA binding. It seems there is a bit more to the story but given the complexity of the phosphorylation events evident in supplemental Figure 7A this will take a very long time to understand.

Fig7 is confusing as visually it suggests that PABP is phosphorylated not TTP. Refining this diagram and supplying a more comprehensive figure legend will aid the reader in understanding the concept.

The statement in the results section "Together, these data demonstrate that phosphorylation of specific serines in TTP's C-terminal IDR is essential for PABPC1 engagement and contributes to mRNA repression, even though phosphorylation does not directly impact TTP-targeted CCR4-NOT-mediated deadenylation." can be misinterpreted as it is not made unambiguously clear what is meant by "mRNA repression" if not CCR4-NOT mediated deadenylation and decay.

Reviewer #3

(Remarks to the Author)

Reviewer #4

(Remarks to the Author)

The new version of the manuscript has been significantly improved, and authors have appropriately replied to my concerns. There are no major concerns with this manuscript in its current form, thus this work is suitable for publication in Nature Communications.

Reviewer #5

(Remarks to the Author)

The reviewers made a concerted effort to address reviewer comments with additional experimentation, explanation and revisions. I have no further constructive comments to offer.

AUTHOR RESPONSE TO REVIEWERS

We thank the reviewers for their enthusiastic and thoughtful feedback. Despite unprecedented disruptions to our operations, we have diligently addressed every comment and performed additional experiments.

Our revised study now includes:

- Measurement of endogenous mRNA levels for three bona fide TTP targets in HEK-293T cells, demonstrating the impact of multivalency in targeted deadenylation.
- Identification of the key phosphoserine residues in TTP that mediate PABPC1 binding, and analysis of how phosphosite mutations affect IL-3 protein expression and endogenous target mRNA levels.
- Cellular verification of TTP interactions with both the NOT module and PABPC1 using co- and re-localization confocal microscopy studies.
- Examination of 14-3-3 engagement and TTP hyperphosphorylation on RNA binding and targeted deadenylation.

Our point- by- point responses appear in blue font, and all manuscript changes are highlighted in dark red.

Reviewer #1 (Remarks to the Author):

The authors of this manuscript focus on analyzing different regions of the human protein tristetraprolin (TTP) that are important for interacting with CCR4-NOT and for eliciting degradation of a labile AU-rich RNA. The authors report two main discoveries. One complements earlier reports that additional regions of TTP are implicated in binding CCR4-NOT; the other is that PABPC1 associates with TTP in a manner dependent on TTP phosphorylation, and this interaction removes PABPC1 from the poly(A) tail, further contributing to RNA decay. Through these studies, the authors propose additional details of TTP-elicited degradation of AU-rich mRNAs.

Given the great importance of understanding the function of RNA-binding proteins through their association with other proteins and post-translational modification, the topic addressed in the current manuscript is timely and significant. Unfortunately, over a dozen articles have reported functional interactions between TTP and CCR4-NOT for the past decade, which reduces the novelty of the authors' findings, although they do uncover additional structural details of the interaction of these proteins. In the last figure, the authors introduce a new protein interacting with TTP physically and functionally, PABPC1; although this finding is certainly exciting and novel, it is far too preliminary. One final concern is that the entire piece is based on biochemical in vitro analysis of recombinant TTP, CCR4-NOT, and PABPC2, and a synthetic labile RNA. To fully support their model, and the impact of the newly identified interactions, the authors need to test endogenous proteins and mRNAs. These concerns are explained below.

We thank the reviewer for these thoughtful insights and agree that demonstrating our biochemical findings in a cellular context will significantly strengthen the manuscript.

Main concerns

1. Figure 1: if the authors wish to connect the functional consequences of the 13 C-terminal residues of TTP and the tryptophan mutations in cells, to the effect of CCR4-NOT on the stability of IL-3 mRNA, they should analyze IL-3 mRNA levels and half-life, not only the levels of secreted IL-3.

Concerning the use of *I/3* mRNA and protein as a TTP target in these experiments, we have reminded readers of the original studies by Stoecklin and his colleagues (PMID: 11486017 and PMID: 10805719), in which *I/3* mRNA was identified in mouse cells as a physiological target of TTP. These authors also used full-length mouse *I/3* mRNA to demonstrate the effect of TTP on destabilizing *I/3* mRNA in co-transfection experiments in NIH 3T3 cells. They also showed the same effect when the *I/3* mRNA 3' UTR was linked to the β -globin mRNA. They also demonstrated the absolute requirement of an AU-rich region in the *I/3* mRNA for TTP-induced destabilization.

We have also cited a previous study by the Blackshear group (PMID: 11279239), in which the effects of TTP and its family members were determined in co-transfection studies in HEK-293 cells. In that paper, they compared *Tnf*, *Csf2* (also known as GM-CSF), and *I/3* transcripts for their potential use in assays of TTP activity in this cell system. In all cases, these transcripts encoded secreted proteins whose mRNAs had previously been identified as physiological TTP targets. Those studies demonstrated the advantages of using *I/3* mRNA as the target in this assay compared to the other two physiological TTP targets. They not only showed an increase in mRNA decay induced by TTP and its family members, but also that this was accompanied by mRNA deadenylation. Additionally, they found that the 32-nucleotide AU-rich region was required for this destabilization effect.

Unfortunately, *I/3* mRNA is not expressed to any appreciable extent in the primary macrophages that are the best cellular source for examining the effects of endogenous WT and mutant TTP on mRNA target decay after inflammatory stimuli. However, we took advantage of earlier data documenting *I/3* mRNA as a TTP target in mouse cells and its deadenylation and decay in response to TTP in co-transfected HEK-293 cells to develop a convenient protein expression assay for the quantitative evaluation of TTP's effects on secreted *I/3*. We were aware that mRNA expression, translation, and secretion could all be affected by TTP, but, based on the previous studies cited above, we were confident about its effects on mRNA decay and deadenylation. It is not practical to create new knock-in mice for the large number of TTP mutants being assayed, and this assay represents a pragmatic approach to testing many new mutants with an anticipated small effect size in a statistically meaningful manner.

In addition to citing this previous literature and extending our analysis beyond secreted IL-3 levels, we have now measured mRNA abundance of three endogenous TTP targets in HEK-293T cells by RT-qPCR (Fig. 4e; Supplementary Fig. 6e). These endogenous decay profiles mirror the modest effects on stability observed for the CNBD deletion and Trp- mutant TTP in our IL-3 HiBiT reporter assay (Fig. 1g), reinforcing that both assays report on the same underlying decay mechanism.

Because the *I/3* transcript in our initial experiments is expressed from a transient transfection and intentionally lacks a regulated promoter, we focused on bona fide targets for mRNA

measurements. We fully agree that testing endogenous mRNAs provides the most physiologically relevant assessment of TTP function, and these new data substantiate our conclusion that removal of the 13 C-terminal residues or mutation of the conserved tryptophans produces only a minor defect in mRNA decay. As we note in the present study, relatively minor effects caused by the removal of the C-terminal 13 residues were previously demonstrated in macrophages derived from knock-in mice with that deletion (PMID: 31036567). However, in that case, primary macrophages were used, and four previously identified bona fide TTP target transcripts were shown to decay more slowly in the mouse macrophages containing the C-terminal TTP deletion: *Tnf*, *Cxcl1*, *Cxcl2* and *Ptgs2* mRNAs, all of which were reasonably well expressed in those cells.

2. The authors rely extensively on in vitro deadenylation assays, which are quick and convenient. However, whether these assays reflect in vivo deadenylation and decay of specific mRNAs in this paradigm, as regulated by these proteins, needs to be tested too. Studying how these interactions affect the stability of TTP targets (at a minimum IL-3 mRNA) in cells is essential to complement the data in Figures 1, 3 and 4.

Many of these issues were addressed in our response to point 1. However, the reviewer may be aware of the first demonstration of TTP-promoted deadenylation, which accompanies and precedes mRNA decay in eukaryotes. This was found fortuitously in bone marrow-derived mouse stromal cells, in which the *Csf2* (GM-CSF) mRNA was identified as a bona fide TTP target (PMID: 10706852). Stromal cells from the TTP KO mice had *Csf2* mRNA that was fully polyadenylated, whereas cells from their WT counterparts had a mixture of fully polyadenylated and deadenylated transcripts. This was documented by RNase H experiments, as well as mRNA decay experiments. Very few other endogenous TTP targets exhibit such well-defined deadenylated mRNA intermediates in the steady state; however, these can be observed using northern blots during mRNA decay for numerous physiological targets.

In addition, we have now performed RT-qPCR measurements of three human endogenous TTP targets (*IER3*, *CXCR4*, and *PIM3*) in HEK-293T cells expressing wild-type or variant TTP proteins (Fig. 4e). These assays complement our in vitro deadenylation data and co-transfection cellular data and reveal that deletion of the CNBD, tryptophan-to-alanine mutations (4xW/A), or combined interface disruptions each produce correspondingly reduced mRNA decay, mirroring the trends observed in our IL-3 HiBiT reporter assays (Figs. 1, 3, 4).

We note that, with the exception of the RNA-binding-deficient C124R mutant, all TTP variants accumulate to higher steady-state levels than wild-type TTP (Supplementary Fig. 6e). Consequently, the decay defects we observe likely underestimate the actual impact of disrupting CCR4-NOT engagement when protein concentrations are equal. Moreover, none of the mutants, aside from C124R, is entirely inactive in promoting CCR4-NOT-mediated deadenylation in vitro, underscoring that partial activity combined with elevated expression may mask even larger functional differences.

Together, these endogenous mRNA decay measurements reinforce our conclusion that multiple, multivalent TTP:CCR4-NOT interactions are required for maximal deadenylation and transcript turnover in cells.

3. The data in Figure 6 are the most novel and potentially exciting, but they open many questions for the model.

Does PABPC1 binding cooperate with CCR4-NOT binding to a target mRNA? do these binding activities synergize? or perhaps compete?

We thank the reviewer for these thought-provoking questions regarding the interplay between PABPC1 and CCR4-NOT in mRNA decay. While our current data establish that TTP phosphorylation enhances PABPC1 binding and promotes PABPC1 displacement from the poly(A) tail (Fig. 6), the precise functional relationship between PABPC1 engagement and CCR4-NOT recruitment remains to be fully elucidated. We offer the following context and hypotheses:

- In *S. pombe*, the PABPC1 ortholog Pab1p can interact directly with the CCR4-NOT complex at sub-stoichiometric ratios without inhibiting deadenylation (PMID: 29932902), suggesting that PABPC1 and CCR4-NOT can occupy the same mRNA tail.
- Given the low cellular concentration of CCR4-NOT (~30 nM; PaxDb) versus abundant PABPC1 (~ μ M range), we speculate that CCR4-NOT is initially recruited by high-affinity RBPs (e.g., TTP), which then, upon phosphorylation, bind PABPC1 to “peel” it from the poly(A) tail. This sequential handoff may facilitate CCR4-NOT access to the tail without direct competition.

Thus, in a **synergistic handoff model**, TTP first recruits CCR4-NOT via multivalent interfaces; subsequent phosphorylation-dependent binding to PABPC1 displaces PABPC1 from the poly(A), clearing the way for CCR4-NOT to engage and deadenylate efficiently. On the other hand, in a **competitive displacement model**, PABPC1 binding by phosphorylated TTP might transiently compete with poly(A) binding, weakening PABPC1’s protection of the tail and thereby indirectly enhancing CCR4-NOT activity.

There is a related regulatory precedent in the case of PAIP2, a translational repressor that binds PABPC1’s RRM2-3 to displace it from poly(A) and stimulate decay (PMID: 11172725). The mechanistic parallel of different RBPs’ ‘peeling off’ PABPC1 to promote decay highlights a potentially conserved strategy.

Discriminating between these models will require kinetic and structural studies of ternary TTP:PABPC1:CCR4-NOT assemblies, as well as cell-based assays that monitor tail length and complex composition over time. We view these questions as exciting avenues for follow-up work and collaborations with laboratories specializing in single-molecule protein/mRNA binding kinetics.

Once again, we appreciate the reviewer’s insightful questions, which have helped us to frame important next steps in unraveling the coordination between PABPC1 displacement and CCR4-NOT-mediated deadenylation.

What specific TTP sites are phosphorylated in this paradigm of displacing PABPC1 from poly(A) tails.

We mapped eight phosphorylated serine residues within TTP's C-terminal intrinsically disordered region (IDR) that are critical for PABPC1 engagement (Supplementary Fig. 9a).

All eight phospho-sites are conserved across mammalian TTP family members and even in more distant orthologs (Supplementary Figs. 1,2, red arrows), supporting a broadly conserved mechanism by which phosphorylation promotes TTP's association with and displacement of PABPC1 from the poly(A) tail.

Furthermore, the interactions invoking PABPC1, physical and functional, need to be studied in vivo in cells, as well as on cellular mRNAs, not only using in vitro binding of recombinant proteins and decay assays.

We thank the reviewer for highlighting the importance of validating TTP:PABPC1 interactions in a cellular context. To address this:

- We transiently expressed wild-type TTP or the phospho-deficient 8xS/A mutant in HEK-293T cells and performed pull-down assays using TTP- specific antibodies. Wild-type TTP robustly co-precipitated endogenous PABPC1, whereas the 8xS/A mutant failed to do so (Supplementary Fig. 9b).
- In parallel, we measured both protein output and mRNA levels of four TTP targets following expression of wild-type or 8xS/A TTP. The 8xS/A mutant showed significantly reduced repression of target protein expression and correspondingly attenuated mRNA decay (Supplementary Fig. 9d,e), despite retaining full activity in our in vitro CCR4-NOT deadenylation assays (Supplementary Fig. 9c). These data support a functional role for TTP phosphorylation in displacing PABPC1 and promoting decay of endogenous transcripts.
- We attempted live-cell imaging of fluorescently tagged TTP and PABPC1 in HEK-293T and HeLa cells, analogous to our NOT-module colocalization studies (Supplementary Fig. 4). However, high levels of endogenous PABPC1 limited our ability to resolve specific TTP:PABPC1 interactions under these conditions.

We agree that further in vivo studies, such as employing an auxin-inducible degron to titrate endogenous PABPC1, would offer deeper mechanistic insight. We look forward to pursuing this exciting future direction in collaboration with laboratories that possess the necessary systems and expertise.

Minor comments

1. The block letters used in some of the figures (black outline, orange fill) would be more visible if displayed in a solid color.

Done.

2. The title, why do the authors use the word 'Repression' instead of the actual process they measure, which is 'Decay'? in fact, they do not actually measure any mRNA, only partial RNAs.

We thank the reviewer for this important observation. Previous studies in TTP-deficient macrophages and fibroblasts have indeed examined how wild-type or mutant TTP influences the decay of full-length endogenous mRNAs in a cellular context. In the present study, in addition to our in vitro deadenylation assays on synthetic RNA substrates, we directly measured full-length mRNA decay in cells by two complementary approaches:

- We use an IL-3 HiBiT assay in HEK-293 cells, which reflects decay of the complete *Irf3* mRNA and its potential translational silencing (repression), not just its 3' end fragment. As discussed above, the effect on full-length *Irf3* mRNA decay and deadenylation had been described previously.
- We measure steady-state levels of three bona fide TTP targets (*IER3*, *CXCR4*, and *PIM3*) by RT-qPCR after expressing TTP or a mutant form of TTP. These data assess the abundance — and by inference, the decay — of native, full-length transcripts (Fig. 4e; Supplementary Figs. 6e, 9e).

Therefore, we chose “Repression” to encompass both translational and decay outcomes that have historically been attributed to TTP. Also, to address the comment by reviewer #4, we have now clarified in the text that the repression we describe is deadenylation-dependent mRNA decay mediated by the CCR4-NOT complex.

Reviewer #2 (Remarks to the Author):

The manuscript by Pekovic et al investigates the TTP mediated RNA decay via its interactions with the CCR4-NOT complex and PABPC1. The authors present a body of work which reveals interesting novel findings with regards to the molecular mechanism of RNA decay mediated by TTP. The authors characterize the low complexity N- and especially C-terminal regions of TTP to be crucial for recruitment of RNA to the NOT complex. In contrast they find that the CCR4-NOT targeting domain of ZFP36 seems to be dispensable for CCR4-NOT mediated RNA deadenylation.

Also, the authors find that phosphorylation of TTP does not affect its RNA decay promoting function which is in contrast to previous models described for the role of TTP phosphorylation. Phosphorylation is however important for TTP interaction with PABPC1 which results in destabilization of RNA. This is an important finding.

The data presented here is interesting and challenges existing models of the molecular mechanisms of TTP mediated RNA decay. However, the manuscript could offer more to demonstrate that the mechanism is physiologically relevant.

We thank the reviewer for their thoughtful summary and for highlighting the novel insights our work provides into TTP-mediated RNA decay. We agree that demonstrating physiological relevance is crucial and have incorporated additional cell-based validation experiments, which also address suggestions from other reviewers.

Specific comments

The authors identify NOT2 and NOT3 as components of CCR4-NOT complex to interact with N terminal low complexity regions of TTP. Multiple proteins are associating with the CCR4-NOT complex with potentially even higher affinities than the interaction described here. An

experiment in a cellular system which demonstrates this interaction in the presence of multiple different competing proteins would strengthen the author's case.

We thank the reviewer for this valuable suggestion. To evaluate TTP's interaction with CCR4–NOT in the context of competing RBP partners, we adapted a re- and co-localization assay (PMID: 38570497) for use in HEK-293T cells (Supplementary Fig. 4). In this assay:

- Fluorescently tagged TTP and the NOT1 subunit of the NOT module co-localize in cytoplasmic foci, confirming their association in live mammalian cells.
- Co-expression of NANOS3, an RBP known to bind NOT1 via a high-affinity α -helical motif (PMID: 24736845), does not disrupt TTP:NOT1 co-localization.

Because NANOS3 binds the same NOT module through a well-defined short helix, our data demonstrate that TTP's multivalent interfaces can outcompete a high-affinity single-motif interaction in cells. This supports a model in which TTP engages multiple CCR4–NOT contacts to secure complex assembly even in the presence of competing RBPs. We agree that further systematic competition experiments, especially involving additional NOT2/NOT3 partners such as PUMILIO, would be informative, and we will pursue these in a dedicated follow-up study.

The authors show that TTP phosphorylation does not inhibit its ability to promote RNA deadenylation. It is known that TTP can be phosphorylated on multiple sites, some sites have been studied in cells and reported to promote TTP stability and to inhibit its ability to promote mRNA decay. (a) How do the phosphorylation events induced in this work relate to the known phosphorylation sites? (b) Which sites are phosphorylated? (c) Can this explain the discrepancy with published work investigating individual phosphorylation sites and the data shown here which seems to induce the phosphorylation of all sites? In a more complex, live cell setting, where only some sites are phosphorylated, this may become relevant.

We thank the reviewer for highlighting the complexity of TTP phosphorylation and its functional consequences. Below, we address points (a-c).

(a) Our purified TTP, expressed in HEK-293T or Sf21 cells, reflects a heterogeneous mix of phosphorylation events occurring under near-physiological conditions. By mass spectrometry of these preparations, we confirm phosphorylation at well-characterized regulatory sites Ser-60 and Ser-186 (Fig. 5b), both previously implicated in TTP stability and mRNA-decay regulation.

(b) Although our sequence coverage was < 70%, thus limiting detection of many C-terminal IDR sites, we observed clear phosphorylation of Ser-60 and Ser-186. Moreover, the pronounced electrophoretic mobility shift caused by mutating eight C-terminal serines to alanine (8xS/A; Supplementary Fig. 9a) strongly indicates that multiple additional IDR residues are phosphorylated in our preparations.

(c) To capture a more uniformly phosphorylated TTP, we treated transfected HEK-293T cells with okadaic acid, yielding "hyperphosphorylated" TTP (Supplementary Fig. 7c). This form exhibits:

- Loss of RNA binding (Supplementary Fig. 7d)

- Reduced interaction with the NOT1 N-terminal CNBD interface (Supplementary Fig. 7f)
- Abolished stimulation of CCR4-NOT-mediated deadenylation (Supplementary Fig. 7e)

These effects are consistent with earlier reports (e.g., PMID: 23644599) showing that phosphorylation within the C-terminal 13 residues impairs NOT1 binding and TTP activity. In contrast, our “basal” phosphorylated TTP (without okadaic acid) retains RNA binding and CCR4-NOT recruitment, explaining why blanket phosphorylation in prior cellular studies led to reduced decay activity.

We agree that in vivo, site-specific phosphorylation likely fine-tunes TTP’s function: selective modification of Ser-60, Ser-186, or individual C-terminal serines may differentially impact RNA binding, CCR4-NOT recruitment, or 14-3-3 engagement. Dissecting these nuanced regulatory layers will be an important focus of future studies.

The authors should, at least, discuss that phosphorylated TTP can be bound by 14-3-3 proteins which can influence TTP function. Could addition of a 14-3-3 protein prevent phosphorylated TTP from promoting deadenylation?

We thank the reviewer for raising the important issue of 14-3-3-mediated regulation of phospho-TTP. To explore whether 14-3-3 binding might antagonize TTP’s deadenylation activity, we performed the following experiments:

14-3-3 binding assays with TTP;

- We observed only sub-stoichiometric binding of 14-3-3 β and 14-3-3 η to recombinant TTP purified from Sf21 or HEK-293T cells (Supplementary Fig. 7b,c).
- Hyperphosphorylated TTP (via okadaic acid treatment) exhibited near-stoichiometric 14-3-3 binding (Supplementary Fig. 7c), but this preparation lacked RNA-binding activity (Supplementary Fig. 7d), precluding direct deadenylation assays with 14-3-3.

Functional assays using ZFP36L2 as a proxy;

- Because ZFP36L2 robustly binds 14-3-3 (Supplementary Fig. 7b) yet remains active in our CCR4-NOT deadenylation assay (Fig. 5c), we tested whether excess 14-3-3 could inhibit ZFP36L2-mediated deadenylation.
- Neither a 10-fold excess of 14-3-3 over ZFP36L2 reduced RNA binding (Supplementary Fig. 7g) nor a 20-fold excess impaired deadenylation stimulation (Supplementary Fig. 7h).

Together, these data suggest that direct competition between 14-3-3 and CCR4-NOT engagement is unlikely to block deadenylation. Instead, we speculate that 14-3-3 may regulate TTP in cells by sequestering it or modulating its phosphorylation dynamics, rather than by directly preventing RNA engagement or CCR4-NOT recruitment.

Might 14-3-3 protein block the TTP interaction with PABPC1?

This is an interesting idea. To test whether 14-3-3 might occlude PABPC1 binding to phosphorylated TTP, we performed competition pull-downs using hyperphosphorylated TTP with both 14-3-3 η and PABPC1:

- Even in the presence of a 4-fold molar excess of 14-3-3 η over phospho-TTP, PABPC1 binding to phospho-TTP was only slightly reduced (<10%) (Supplementary Fig. 9f).
- Conversely, a 2-fold molar excess of PABPC1 over phospho-TTP did not significantly diminish 14-3-3 η association.

These results indicate that 14-3-3 and PABPC1 can simultaneously engage phosphorylated TTP, likely facilitated by the flexibility of TTP's intrinsically disordered regions.

The authors describe the interaction of phosphorylated TTP with PABPC1 as displacing PABPC1 to promote deadenylation. Identifying the relevant phosphorylation sites that mediate this interaction would add value to the study.

This suggestion was also made by reviewers 1 and 4. As detailed in our responses to their comments, we have now mapped eight serine residues within TTP's C-terminal intrinsically disordered region that are critical for phospho-dependent PABPC1 binding (Supplementary Fig. 9a). Alanine substitution of these sites (8xS/A) abolishes PABPC1 interaction both in vitro and in HEK-293T pull-downs (Supplementary Fig. 9a,b), significantly attenuates repression of our IL-3 HiBiT reporter (Supplementary Fig. 9d), and reduces decay of three endogenous TTP targets by RT-qPCR (Supplementary Fig. 9e). Importantly, the 8xS/A mutant retains full activity in stimulating CCR4-NOT-mediated deadenylation (Supplementary Fig. 9c), confirming that these phosphosites specifically govern PABPC1 engagement and its role in promoting mRNA decay.

In Figure 1F the authors use a HEK-293 transduction system to investigate the role of modified TTPs on the production of IL3. It is unclear whether altered IL3 production is due to changes in mRNA deadenylation.

We thank the reviewer for this important point, which Reviewer 1 also raised. As described in our responses to their main comments 1 and 2, we have now measured endogenous mRNA levels of three bona fide TTP targets in HEK-293T cells by RT-qPCR (Fig. 4e; Supplementary Fig. 6e). These data show that the derepression of IL-3 protein production in our HiBiT assay mirrors the changes in mRNA abundance for native targets, consistent with altered IL-3 production reflecting differences in mRNA deadenylation and decay rather than downstream translational effects. In addition, as we described in detail in our response to Reviewer 1, we have now referred to previous studies documenting the behavior of *Il3* mRNA in mouse cells by Stoecklin and colleagues, as well as earlier studies by the Blackshear group showing the ability of TTP to promote *Il3* mRNA deadenylation and decay in transfected HEK-293 cells, in a manner entirely dependent on the presence of an AU-rich region in the 3' UTR.

Also, it is worth clarifying whether the HEK-293 cells express any of the TTP family proteins which might also affect the assays.

We thank the reviewer for raising this point. Published studies report that HEK-293 cells express minimal to undetectable levels of endogenous TTP family members (e.g., TTP: see Fig. 5c in PMID 17030620; ZFP36L1: see Fig. 8c in PMID 34774847; ZFP36L2: see Fig. 1a in PMID 29426877). Moreover, our experiments employ CMV-driven overexpression of TTP constructs, resulting in protein levels that far exceed any residual paralog expression. Thus, any contribution from endogenous TTP family proteins in our assays is negligible.

Minor comments:

Line 125 please clarify what the components of the fully reconstituted human CCR4NOT complex include, as it is unclear what its composition is with respect to NOT4 from Ref 51.

We apologize for the lack of clarity. The fully reconstituted human CCR4-NOT complex used in this study comprises all core subunits: NOT1, NOT2, NOT3, NOT6, NOT7, and NOT9, as well as subunits NOT10:11. It does not include the E3 ligase NOT4, which does not stably associate with the rest of the CCR4-NOT in mammals. Additionally, Fig. 2a now explicitly labels each subunit in the schematic of the full complex.

Line 159 the NOT module referred to appears to consist of NOT2 and NOT3 and also the HEAT domain of NOT1? Please clarify here as in the paragraph multiple HEAT domains are referred to.

This is correct. We have clarified the composition as follows: 'We also discovered a previously unreported stoichiometric interaction between TTP and the NOT module, which comprises the C-terminal Spa2 Homology Domain (SHD) together with NOT2 and NOT3, thereby defining three distinct CCR4-NOT binding interfaces.'

Line 180 it is not clear how the authors estimate significance when they claim that the TZF domain significantly enhanced targeted deadenylation activity. Does the TZF domain of TTP used here in these experiments actually bind to RNA? Or does it bind RNA but just is not recruiting it to the CCR4-NOT complex? This is not completely clear.

The TZF domain is the well-characterized tandem zinc finger RNA-binding module of TTP, which binds synthetic AU-rich RNAs with low nanomolar affinity. In pulldown experiments (Fig. 2d), the TZF domain alone does not recruit CCR4-NOT, confirming that its role is strictly RNA binding. By contrast, when fused to the N- or C-terminal IDRs, neither of which binds to RNA, the TZF domain directs these IDRs to RNA, allowing us to assess how each IDR contributes to CCR4-NOT engagement and deadenylation. When the TZF domain is mutated, the IDRs do not contribute appreciably to CCR4-NOT engagement and deadenylation on target RNA. Likewise, on mock substrates that are not recognized by the TZF domain, TZF-fused constructs display no activity above background. From this, we conclude that the TZF domain makes a significant contribution to targeted deadenylation.

Line 207 please indicate that the full length TTP is 1-326 for more clarity.

Done.

Line 209 it is not clear what the authors are referring to. Is it fragment 1-233? Please refer to the fragments as described in the figure. If it is fragment 1-233, it seems to me that the conclusion of the experiment is not correct.

We apologize for the confusion around fragment numbering. In the lines in question, we are indeed referring to fragment 1-248 (as defined in Fig. 3), not 1-223. The apparent discrepancy arises because the gel in Fig. 3d reuses samples from Supplementary Fig. 6a, where higher amounts were loaded to visualize weaker interactions (e.g., fragment 102–248). This makes the modest reduction in NOT module binding for fragment 1-248 less noticeable in that panel, but the same trend is evident when comparing fragments 102-248 versus 102-326 and fragment 102-223 (Supplementary Fig. 6a), as well as in the deletion construct Δ 249-274 (Supplementary Fig. 6c). Together, these data consistently map the C-terminal NOT module interface to residues 224-274.

Line 259 Can the authors please clarify whether they mean phosphatase inhibitors when they write "...phosphatase-treated purified proteins to bacterially expressed non phosphorylated controls".

Here, we treated our HEK-293T- or Sf21-expressed TTP, which carries native eukaryotic phosphorylation, with λ -protein phosphatase to remove those phosphate groups. We then compared the dephosphorylated proteins to bacterially expressed TTP, which, lacking the eukaryotic modification machinery, serves as our control for non-phosphorylated proteins.

Line 265 the authors state that phosphorylation did not affect TTPs RNA binding. However actual binding to RNA by TTP is not shown as far as I can tell.

We appreciate the reviewer pointing this out. While our initial conclusion was based on the equivalent deadenylation activity of phosphorylated versus non-phosphorylated TTP (Fig. 5c), we recognize that direct RNA-binding data are necessary. To clarify:

- We now show in Supplementary Fig. 7d that TTP purified from okadaic acid–treated HEK-293T cells (hyperphosphorylated) exhibits no detectable RNA-binding in our electrophoretic mobility shift assays.
- In contrast, phosphorylated ZFP36L2 retains full RNA-binding activity (Supplementary Fig. 7g), consistent with its deadenylation stimulation in Fig. 5c.

It was previously demonstrated that TTP expressed in HEK-293 cells (and thus basally phosphorylated) binds an ARE probe significantly less well than the same protein following dephosphorylation, even in the absence of okadaic acid or PMA (PMID: 11546803).

Line 310-312, clarify that the deadenylation assays contain the CNOT complex as the "reactions containing only PABPC1" was slightly confusing.

Done.

Figure Legends 1G and 3E and 4D. Please clarify what each dot represents.

Done.

Please also state in all figure legends how many times each representative experiment has been performed.

Done.

Please also indicate what each figure panel represents. If Fig.1B shows a Coomassie gel, please indicate that. Same for Panels like 1C.

Done.

Please explain the rationale for why in some experiments HEK-293 cells are used and in other experiments Sf21 insect cells are used.

Two complementary needs drove the choice of expression system in our study.

HEK-293 cells provide a native mammalian cellular context in which TTP (and its variants) is phosphorylated, interacts with endogenous partners (e.g., PABPC1, 14-3-3), and engages the cellular decay machinery. All co-IPs, co-localization microscopy, HiBiT reporter assays, and RT-qPCR measurements of endogenous mRNAs were performed in HEK-293 or HEK-293T cells to capture these physiologically relevant interactions and modifications.

Sf21 insect cells allow robust, scalable production of recombinant TTP (and its family members) and some of the CCR4-NOT subunits with proper eukaryotic folding and minimal bacterial contaminants. This is critical for our in vitro biochemical assays, such as deadenylation reactions and pull-downs, that require milligram quantities of pure, homogeneous proteins. Furthermore, by expressing TTP in Sf21 cells, we obtain high-yield preparations with native phosphorylation patterns (when untreated) or allow controlled dephosphorylation (using λ -phosphatase) for biochemical studies.

Reviewer #3 (Remarks to the Author):

We thank the reviewer and their co-reviewer for their thoughtful evaluation and for participating in the journal's initiative to support peer review training and early-career recognition. Your time and insights have substantially improved our work.

Reviewer #4 (Remarks to the Author):

In this manuscript, authors investigated how tristetraprolin (TTP) interacts with CCR4-NOT and PABPC1 to control mRNA stability. Using purified proteins and in vitro assays, authors discovered that TTP engages CCR4-NOT through multiple interaction sites and promotes its deadenylation activity on mRNAs containing AU-rich elements. Authors also observed that phosphorylation of TTP did not affect its interaction with CCR4-NOT or its deadenylation activity, but was essential for TTP's binding to PABPC1.

This manuscript is an interesting piece of work. It is well-written and logically presented. However, data does not seem to fully support the authors' main conclusions. Some issues should be clearly addressed before publication.

We thank the reviewer for their positive assessment of our study and for highlighting areas for improvement. Below, we address each concern in detail.

Specific comments:

Regarding the model proposed, authors say "TTP promotes the processive deadenylation activity of CCR4-NOT on mRNAs containing AU-rich elements, with phosphorylation-dependent interactions with PABPC1 potentially enhancing deadenylation and promoting regulated mRNA decay". More experimental results should be included to confirm that interactions of phosphorylated TTP with PABPC1 enhance deadenylation and promote regulated mRNA decay.

As detailed in our responses to Reviewers 1 and 2, we have now provided direct evidence that TTP phosphorylation promotes PABPC1 recruitment and can contribute to regulated mRNA decay in cells:

- We mapped eight serine residues in TTP's C-terminal IDR essential for PABPC1 binding (Supplementary Fig. 9a).
- The phospho-deficient 8xS/A TTP fails to co-IP endogenous PABPC1 (Supplementary Fig. 9b).
- The 8xS/A mutant exhibits attenuated repression of the IL-3 HiBiT reporter (Supplementary Fig. 9d) and reduced decay of endogenous TTP targets, as determined by RT-qPCR (Supplementary Fig. 9e).

Together, these data support a model in which the phosphorylation-dependent recruitment of PABPC1 by TTP contributes to CCR4-NOT-mediated deadenylation and regulated mRNA decay in a cellular context.

Also, it would be interesting to show how the proposed mechanism of controlling mRNA decay responds to inflammation in the cell. In fact, the manuscript does not present in cellulo experiments and this would reinforce conclusions about the cellular role of TTP in the control of mRNA decay in response to cellular stimuli, such as inflammation.

The involvement of TTP itself in the regulation of innate immunity dates back many years, from the initial description of the inflammatory phenotype seen in the KO mice (PMID: 8630730), to the identification of TTP as an RNA binding and destabilizing protein that could bind directly to AU-rich elements of the *Tnf* mRNA (PMID: 9703499), to the elucidation of deadenylation as the first step in the stimulation of TTP-mediated mRNA decay (PMID: 10706852). Many follow-up studies by the Blackshear group and others have contributed to various aspects of this work. However, many aspects remain to be further elucidated, such as the synergistic contributions of the other family members in promoting both mRNA decay and the inflammatory phenotype in mice (PMID: 37903626). Complicating this effort are the significantly different mouse phenotypes observed when ZFP36L1 and ZFP36L2 are

knocked out, involving physiological systems such as embryo development and hematopoiesis that are not typically associated with inflammation.

Several structure-function studies of TTP in the intact mouse have been performed, but most have disrupted major functional elements. For example, mice with a non-RNA-binding mutation of TTP phenocopy the complete knockout (PMID: 29203639), whereas mice in which the C-terminal CNBD domain has been removed exhibit a milder but still detectable inflammatory phenotype (PMID: 31036567). However, the results of more 'granular' loss-of-function mutations would be expected to be subtle, in part because of the degrees of cross-reactivity among the family members (PMID: 37903626). They may not be feasible due to practical considerations.

On the other hand, gain-of-function mouse knock-ins have also been performed to test the concept that increasing TTP activity or amount might have potential therapeutic benefits. These include a mouse in which an AU-rich region of the mRNA 3' UTR has been removed, which can protect against many models of inflammatory diseases (reviewed in PMID: 35525391). Another approach has been taken by the Clark group, which mutated two of the 14-3-3-interacting serines to alanines and developed a mouse that also exhibited an anti-inflammatory phenotype (PMID: 26002976, 28265004).

Each of these models has taken many years to develop and phenotype, and this approach is made more challenging when loss-of-function mutations are expected to produce more subtle phenotypes, as discussed in cell transfection studies. Developing new knock-in mice is not a practical way to study these subtle mutations, given the time and resources required for this approach. Conducting these *in vivo* experiments would represent a substantial expansion of the study that cannot be supported within the current framework, especially given our current funding and personnel constraints within the organizational setting.

One view is that elucidation of some of the finer loss of function mutations may be accomplished in organisms in which only a single TTP family member is expressed, and in which the effects on mRNA decay are very similar to those observed here: an example includes *Dictyostelium discoideum*, which expresses only one TTP family member, which has the same functional domains as described here for the human proteins (PMID: 34718768).

Although we do not present *in vivo* mouse experiments here, our manuscript does include several cellular assays that directly address TTP function in a cellular context. For example, we employ a HiBiT reporter assay in transfected cells to quantify TTP's effect on IL-3 production, perform co-localization microscopy to visualize TTP association with CCR4-NOT, and analyze endogenous mRNA decay targets by qPCR following TTP expression. These *in cellulo* experiments already demonstrate that our biochemical observations translate into a cellular setting. Extending this work to incorporate inflammatory stimuli in primary immune cells or animal models would, as suggested, be highly informative; however, these studies would require significant additional resources and specialized expertise and would expand the scope of this manuscript far beyond our current aims. We trust the reviewer will agree that the existing cellular data provide a robust foundation for our mechanistic conclusions.

Minor comment:

Authors should specify the type of mRNA repression.

We thank the reviewer for highlighting this omission. We have now clarified throughout the text that the repression we describe is deadenylation- dependent mRNA decay mediated by the CCR4-NOT complex. Specifically, the first instance of “mRNA repression” has been defined as “CCR4-NOT- mediated deadenylation- dependent decay of target transcripts leading to silencing of translational output” making clear that TTP promotes shortening of the poly(A) tail, which subsequently triggers decapping and 5'-to-3' exonucleolytic degradation.

Reviewer #5 (Remarks to the Author):

Pekovic et al investigate determinants required for interactions between CCR4-NOT, PABPC1 and TTP in a manuscript entitled ‘Multivalent Interactions with CCR4-NOT and PABPC1 Determine mRNA Repression Efficiency by Tristetraprolin’. In this work, they primarily rely on reconstitution biochemistry, alphafold predictions, and RNA decay assays to investigate multivalent interactions between TTP and CCR4-NOT which serves to recruit deadenylase activity to model RNA substrates. Of note, phosphorylation of TTP was thought to promote recruitment of CCR4-NOT to enhance deadenylation, but the authors found that phosphorylation had no effect on interactions with CCR4-NOT. Instead, phosphorylation appears to enhance TTP association with the polyA binding protein PABPC1, an unexpected but very nice result which explains how phosphorylation may promote deadenylation.

The biochemical assays are well presented, especially with respect to analysis of RNA mixtures (containing mutations in recognition motifs) as the authors compared rates of decay in assays with WT and mutant uniquely labeled to enable analysis of the control decay substrate in the same reaction.

This is a clean and concise study that illuminates a complex network of interactions that underlie recruitment of factors to elicit deadenylation and promote decay. There are no major concerns with this manuscript in its current form, and only a few minor suggestions are offered.

We thank the reviewer for their generous and thoughtful assessment of our work. Below, we address the minor suggestions raised.

In the abstract, the authors state: ‘Here, we investigated how TTP interacts with CCR4-NOT and PABPC1 to control mRNA stability.’ Given that most of the work is done on synthetic RNAs and not on mRNA per se, the authors might restate ‘control mRNA stability’ to ‘promote RNA deadenylation’.

To reinforce that our findings extend beyond synthetic substrates, we have now included RT-qPCR measurements of steady-state mRNA levels for three bona fide TTP targets following overexpression of wild-type and phosphomutant TTP in HEK-293T cells (Fig. 4e; Supplementary Figs. 6e, 9e). Detailed descriptions of these mRNA stability assays have been incorporated into the Results section and are also provided in our responses to Reviewer 1.

One line 248 the sentence starts with 'This study addressed...' and they then reference 7 references at the end of the sentence. What does 'This study' refer to?

We apologize for the ambiguity. This was rephrased.

The authors use the terms 'completely abolished' on lines 286 and 303. Abolished is an absolute term, it should not be prefaced with completely or any other qualifier.

Done.

Consider removing the eyeballs on CNOT6/7 in Fig 2a, maybe ok for a review, but seems a bit too cute for a primary research manuscript.

Done.

Consider adding schematic/cartoon indicating a model/role for phosphorylation in the context of these interactions.

Thank you for this suggestion. We have updated Fig. 7 to include a schematic that highlights our proposed model for how the phosphorylation of TTP (denoted by red letters "P") regulates its interaction with PABPC1 and the CCR4-NOT complex.

AUTHOR RESPONSE TO REVIEWERS

Our point- by- point responses appear in blue font, and all manuscript changes are highlighted in dark red.

Reviewer #2 (Remarks to the Author):

The authors provide a comprehensive revision which addresses the points raised. Overall, this is a significant body of work which shows how phosphorylated TTP can displace PABP and promote deadenylation by CNOT. There is still some scope to offer a bit more clarity. This could be to place the work in the context of other phosphorylation events which are outside of 249-313, especially those 60 and 186. Related to this is the observation that chemical induction of hyperphosphorylation of TTP results in loss of RNA binding. It seems there is a bit more to the story but given the complexity of the phosphorylation events evident in supplemental Figure 7A this will take a very long time to understand.

We thank the reviewer for their thoughtful feedback and highlighting the complexity of TTP phosphorylation and its functional consequences.

Fig7 is confusing as visually it suggests that PABP is phosphorylated not TTP. Refining this diagram and supplying a more comprehensive figure legend will aid the reader in understanding the concept.

We have now refined both the figure and legend for the proposed model.

The figure legend now states:

“Schematic model illustrating how TTP exploits multivalent interactions with two key regulators of mRNA fate — CCR4-NOT and PABPC1. Short segments within TTP’s intrinsically disordered C-terminal region (red rectangles) serve as docking sites that scaffold these factors and direct rapid, transcript-specific deadenylation. Phosphorylation of TTP (red “lollipop” symbols) promotes high-affinity binding to PABPC1, displacing PABPC1 from the poly(A) tail and thereby clearing access for CCR4-NOT to engage the tail and catalyze efficient deadenylation.”

The statement in the results section “Together, these data demonstrate that phosphorylation of specific serines in TTP’s C-terminal IDR is essential for PABPC1 engagement and contributes to mRNA repression, even though phosphorylation does not directly impact TTP-targeted CCR4-NOT-mediated deadenylation.” can be misinterpreted as it is not made unambiguously clear what is meant by “mRNA repression” if not CCR4-NOT mediated deadenylation and decay.

We rephrased the statement to specify the two layers of mRNA regulation:

“Taken together, our data show that phosphorylation of specific serines in TTP’s C-terminal IDR is required for productive engagement of PABPC1 and thereby lowers both mRNA stability and translational output. Importantly, these phosphorylation events do not measurably alter the efficiency of TTP-directed CCR4-NOT-mediated deadenylation itself.”